# Deriving Causal Order from Single-Variable Interventions: Guarantees & Algorithm

**Mathieu Chevalley**[1,2†]    **Patrick Schwab**[1]    **Arash Mehrjou**[1,3†]

[1]GSK.ai    [2]ETH Zürich    [3] MPI for Intelligent Systems

## Abstract

Targeted and uniform interventions to a system are crucial for unveiling causal relationships. While several methods have been developed to leverage interventional data for causal structure learning, their practical application in real-world scenarios often remains challenging. Recent benchmark studies have highlighted these difficulties, even when large numbers of single-variable intervention samples are available. In this work, we demonstrate, both theoretically and empirically, that such datasets contain a wealth of causal information that can be effectively extracted under realistic assumptions about the data distribution. More specifically, we introduce a novel variant of *interventional faithfulness*, which relies on comparisons between the marginal distributions of each variable across observational and interventional settings, and we introduce a score on *causal orders*. Under this assumption, we are able to prove strong theoretical guarantees on the optimum of our score that also hold for large-scale settings. To empirically verify our theory, we introduce Intersort, an algorithm designed to infer the causal order from datasets containing large numbers of single-variable interventions by approximately optimizing our score. Intersort outperforms baselines (GIES, DCDI, PC and EASE) on almost all simulated data settings replicating common benchmarks in the field. Our proposed novel approach to modeling interventional datasets thus offers a promising avenue for advancing causal inference, highlighting significant potential for further enhancements under realistic assumptions.

## 1 Introduction

Causal structure learning is pivotal for understanding complex systems, aiming to discover causal relationships from data. This field spans disciplines such as biology (Meinshausen et al., 2016; Yu et al., 2004), medicine (Farmer et al., 2018; Joffe et al., 2012; Feuerriegel et al., 2024), and social sciences Baum-Snow & Ferreira (2015); Imbens & Rubin (2015), where causal insights drive informed decision-making and deepen our understanding of underlying processes. Traditionally, causal discovery has relied heavily on observational data, primarily because expansive interventional experiments are often impractical or costly in many application domains. The inherent limitations of observational data necessitate assumptions about data distribution to ensure identifiability beyond the Markov equivalence class (Spirtes et al., 2000; Shimizu et al., 2006; Hoyer et al., 2008). However, the emergence of large-scale interventional data, especially in single-cell transcriptomics data (Replogle et al., 2022; Datlinger et al., 2017; 2021; Dixit et al., 2016), introduces both new opportunities and challenges.

Interventional data, characterized by targeted alterations to a system, offers a unique perspective for causal discovery. These controlled interventions reveal causal mechanisms often obscured in observational studies. As such, many causal discovery methods using interventional data have been proposed (Brouillard et al., 2020; Lorch et al., 2022; Lopez et al., 2022; Hauser & Bühlmann, 2012). However, their practicality remains challenging. For example, recent benchmarks in the gene network domain (Chevalley et al., 2023; 2022) demonstrate that simpler models based on straightforward statistical comparisons between observational and interventional distribution outperforms established causal discovery methods, particularly when many single-variable interventions are

---

[†] Correspondence to `m.chevalley97@gmail.com`, `arash@distantvantagepoint.com`

available. The motivation for this research thus stemmed from these recent advances in applying causal discovery techniques to gene network inference. Our methodology, including the use of the Wasserstein distance, aligns with evaluation metrics for predicted graphs as established in the benchmark by Chevalley et al. (2022). Additionally, a component of our algorithm builds upon and formalizes approaches from leading models in gene network inference, such as those highlighted in the CausalBench challenge (Kowiel et al., 2023; Nazaret & Hong, 2023; Deng & Guan, 2023; Chevalley et al., 2023). This work not only addresses a critical domain of application but also enhances the theoretical framework surrounding contemporary advancements in the field (Yu et al., 2004; Chai et al., 2014; Akers & Murali, 2021; Hu et al., 2020).

Here, we study further the potential of single-variable interventional data for causal structure discovery, which we reduce to finding the *causal order* (topological order) of the variables. Even though the causal order represents a superset of the true causal relationships, it is useful *per se*. Intuitively and informally, correlations are more likely to be causations if they follow the causal order. More practically, it can for example be used to guide experimental design, as it greatly reduces the hypothesis space. As a specific example, in biology, a task of interest is to select pairs of genes to intervene on, which is a space much larger than single gene interventions. For a target gene that let us assume lies in the middle of the causal order, the number of candidate double interventions goes from $\binom{d}{2}$ down to $\binom{\frac{d}{2}}{2}$, where $d$ is the number of variables.

To find the causal order, we introduce a data-based score to rank candidate causal orders, and we establish theoretical guarantees on its optimum, providing upper-bounds on the expected error depending on the graph density and on the probability for a variable to be intervened on, that also hold for large scale settings. Crucially, our approach makes light assumptions on the data distributions, requiring only what we call $\epsilon$-*interventional faithfulness*, which characterizes the strength of changes in marginal distributions between observational and interventional distributions. We then introduce a novel algorithm called INTERSORT that leverages this assumption to derive the causal order.

Our empirical evaluations on diverse simulated datasets (linear, random Fourier features (Lorch et al., 2022), neural network (Nazaret et al., 2023; Brouillard et al., 2020) and single cell (Dibaeinia & Sinha, 2020) with various noise distributions), which are close replications of real-world systems and follow common practice in the field, confirm our theoretical results. The efficacy of our approach in accurately determining the causal order from interventional data signifies a notable advancement in causal inference. INTERSORT outperforms the four baselines, namely PC (Spirtes et al., 2000), GIES (Hauser & Bühlmann, 2012), DCDI (Brouillard et al., 2020) and EASE (Gnecco et al., 2021) on almost all settings. It is also robust to data normalization (Reisach et al., 2021), which is a failure mode of many recently proposed continuous optimization causal discovery methods. Moreover, our empirical results show that Intersort makes more efficient use of interventional information compared to existing approaches which is a critical advantage especially in domains where interventions are costly and sometimes impossible to perform. This suggests that the task of identifying causal structures with many single-variable interventions may be more feasible than previously believed. Moreover, our method demonstrates similar performance and robustness across different types of simulated data, emphasizing its reliability and versatility.

Our method relies on $\epsilon$-*interventional faithfulness* which is a lighter version of many of the existing assumptions in causal discovery literature, while still providing theoretical guarantees. Realistic assumptions on the data distributions are crucial, as in many application domains such as gene expression data, the validity of common assumptions in the field may be unverifiable or known to not hold. Furthermore, many methods catastrophically fail when their assumptions do not hold, rendering them inapplicable (Montagna et al., 2023a; Heinze-Deml et al., 2018).

## 2 RELATED WORK

**Causal ordering** Even though the causal order does not contain the full causal information as it does not uniquely identify the causal graph, it can subsequently be used with for example penalized regression techniques to recover the edges (Bühlmann et al., 2014; Shimizu et al., 2011). Also, the correct causal order is useful in itself as a fully connected graph can be constructed from it which describes the interventional distributions (Peters & Bühlmann, 2015; Bühlmann et al., 2014). More recently, following the discovery of Reisach et al. (2021) that identified an artifact in simulated datasets where sorting variables by variance appears to recover the causal order under certain conditions, many algorithms to recover the causal order from observational data have been proposed,

for example via score matching (Rolland et al., 2022; Montagna et al., 2023a;b). To our knowledge, another work aiming to infer the causal order from interventions comes from Tian & Pearl (2001), which builds on seminal works such as Cooper & Yoo (1999); Spirtes et al. (2000). Tian & Pearl (2001) focus on causal discovery from passive interventions, where shifts in distributions arise naturally. Our work brings two major contributions: first, we propose a score-based approach, in contrast to the rule-based algorithm of Tian & Pearl (2001), which opens the door for a large class of optimization tools and offers better scalability, and second, we provide extended theoretical results in terms of upper-bounding the expected error of our algorithm, especially in cases where only a subset of the variables is intervened on whereas Tian & Pearl (2001) argues mainly about the recovery of the true causal order when "all" variables are intervened. Adjacently, methods like DirectLiNGAM (Shimizu et al., 2011) infer causal orderings from observational data under non-Gaussianity assumptions. As such, they score orderings based on fitted parameters, making it sensitive to distributional assumptions. By leveraging interventional data, our approach relaxes distributional assumptions and provides a more robust framework for causal ordering. Eberhardt et al. (2005; 2006) explored the theoretical bounds on the number of experiments required for full causal structure identification. Our work complements these findings by focusing on deriving causal orderings from single-variable interventions under light assumptions, which can be more practical in large-scale applications.

## 3 CAUSAL ORDER: DEFINITIONS

We here introduce our framework for causality following Pearl (2009) and we follow the notations of Peters et al. (2017). We also present the definition of a causal order as well as a divergence to measure how faithfully a causal order recalls the causal information.

**Statistical metric** Let $(M, d)$ be a metric space, and let $\mathcal{P}(M)$ be the set of probability measures over $M$. We define $D : \mathcal{P}(M) \times \mathcal{P}(M) \to [0, \infty)$ as a distance between probability measures on $M$.

**Distribution** We consider a set of $d$ random variables $X = (X_1, ..., X_d)$ indexed by $V = \{1, ..., d\}$, with associated joint distribution $P_X$. We denote the marginal distribution of each random variable as $P_{X_i}$, $i \in V$.

**Causal Graph** We write $\mathcal{G} = (V, E)$ a DAG between variables $v \in V$, $E \subseteq V^2$, such that $(v, v) \notin E$ for all $v \in V$. We write $A^{\mathcal{G}}$ the $d \times d$ adjacency matrix, where $A^{\mathcal{G}}(i, j) = 1 \iff (i, j) \in E$. We note $\mathbf{Pa}_j$ the parents of $j$, where $i$ is a parent if $(i, j) \in E$. We note $\mathbf{DE}_{\mathcal{G}}^i$ the descendants of $i$ in graph $\mathcal{G}$. $j$ is a descendant of $i$ if there is a directed path from $i$ to $j$. We note $\mathbf{AN}_{\mathcal{G}}^i$ the ancestors of $i$ in graph $\mathcal{G}$. $j$ is an ancestor of $i$ if there is a directed path from $j$ to $i$.

**Causal Order** Given a causal graph $\mathcal{G}$, we call a permutation of the variables:

$$\pi : \{1, ..., d\} \to \{1, ..., d\},$$

a causal ordering of the variables in $\mathcal{G}$ if it satisfies $\pi(i) < \pi(j)$ if $j \in \mathbf{DE}_{\mathcal{G}}^i$ (Peters et al., 2017). A permutation is a bijective mapping of the indices to new indices.

For a causal graph, a causal ordering always exists, but may not be unique. We note the set of causal orderings $\Pi^*$, and any member of $\Pi^*$ as $\pi^*$. We can rearrange the variables according to a permutation $\pi$, and we write the new associate adjacency matrix $A_{\pi}^{\mathcal{G}}$, where $A_{\pi}^{\mathcal{G}}$ is upper triangular if $\pi \in \Pi^*$. Lastly, we define $\overline{A_{\pi}^{\mathcal{G}}} = A_{\pi}^{\mathcal{G}} + A_{\pi}^{\mathcal{G}^2} + ... + A_{\pi}^{\mathcal{G}^{d-1}}$ the transitive closure of the adjacency matrix. It is straightforward to show that $\overline{A_{\pi^*}^{\mathcal{G}}}(i, j) > 0$ if and only if there is a directed path from $i$ to $j$ in $\mathcal{G}$. Furthermore, there may be a total causal effect* of $i$ on $j$ only if $\overline{A_{\pi^*}^{\mathcal{G}}}(i, j) > 0$. $\overline{A_{\pi^*}^{\mathcal{G}}}$ is also upper-triangular.

**Top Order Divergence $D_{top}$** To measure the discrepancy between a permutation $\pi$ and a graph $\mathcal{G}$, we use the top order divergence (Rolland et al., 2022). The top order divergence $D_{top} : (\{1, ..., d\}, \{1, ..., d\} \times \{1, ..., d\}) \times (\{1, ..., d\} \to \{1, ..., d\}) \to \mathbb{N}_0$ is defined as:

$$D_{top}(\mathcal{G}, \pi) = \sum_{\pi(i) > \pi(j)} A^{\mathcal{G}}(i, j).$$

$D_{top}$ counts the number of edges that cannot be recovered given a topological order $\pi$, or said differently, the number of false negative edges of the fully connected DAG corresponding to $\pi$.

---

*There is a total causal effect of $i$ on $j$ if there exists some $\tilde{N}$ such that $X_i \not\perp\!\!\!\perp X_j$ in $P_X^{C, do(X_k := \tilde{N}))}$ (Definition 6.12 in Peters et al. (2017)).

As such, it represents a lower bound on the structural hamming distance (SHD) achievable when constrained to a particular topological order.

*Observation* 1. The divergence $D_{top}$ can also be written as $D_{top}(\mathcal{G}, \pi) = \sum_{i<j} A_\pi^\mathcal{G}(i,j)$.

**Lemma 1.** *Let $\pi^* \in \Pi^*$ be a causal ordering for $\mathcal{G}$. Then $D_{top}(\mathcal{G}, \pi^*) = 0$. (Rolland et al., 2022)*

**SCMs and interventions**   An acyclic structural causal model (SCM) $\mathcal{C} = (\mathbf{S}, P_N)$ is determined by a collection of $d$ assignments $s_j \in \mathbf{S}, \mid \mathbf{S} \mid = d$, such that for each assignment $s_j$ defines $X_j$

$$X_j := f_j(X_{pa_j}, N_j),$$

where $\mathbf{Pa}_j \in V \setminus j$ is the set of variables on which $X_j$ directly depends on and $P_N$ is a joint distribution over the noise variables $N = (N_1, ..., N_d)$ that is jointly independent (Peters et al., 2017). The associated graph $\mathcal{G}$ that connects variables to their parents is assumed to be acyclic.

**Distribution with Intervention $P_X^{(C, do(X_k := \tilde{N}_k))}$**   An intervention is then defined as a replacement of a subset of the structural assignments of an SCM $\mathcal{C}$, such that no cycle is created. In this work, we consider intervention on a single variable, where the structural assignment is replaced by a new random variable independent of the parents. We denote the new distribution entailed by an intervention $P_X^{\mathcal{C}, do(X_k := \tilde{N}_k)}$. We denote $\mathcal{I} \subseteq \{1, ...d\}$ the set of nodes that are intervened on. We also assume that for each intervention, we have access to the interventional distribution $P_X^{\mathcal{C}, do(X_k := \tilde{N}_k)}, k \in \mathcal{I}$. The set of interventional random variables is denoted as $\tilde{N} = \bigcup_{k \in \mathcal{I}} \{\tilde{N}_k\}$.

## 4   $\epsilon$-INTERVENTIONAL FAITHFULNESS

We here introduce the main assumption of our work that we name *interventional faithfulness*. Akin to the common assumption of faithfulness of a joint distribution with respect to the true graph, which assumes that all d-separations in the graph are reflected by conditional independences in the entailed distribution, $\epsilon$-interventional faithfulness assumes that all paths in the graphs are revealed by changes in distribution under intervention above a significance threshold $\epsilon$. This assumption is also similar to how Mooij et al. (2016) define causality as the existence of interventions inducing a change in distribution. The main difference with comparable assumptions made in the literature is that this definition depends on a statistical distance and significance threshold, instead of an oracle of the statistical dependence. We formalize our assumption as follows:

**Definition 1.** Given the distributions $P_X^{\mathcal{C}, (\emptyset)}$ and $P_X^{\mathcal{C}, do(X_k := \tilde{N}_k)}, \forall k \in \mathcal{I}$, we say that the tuple $(\tilde{N}, \mathcal{C})$ is $\epsilon$-*interventionally faithful* to the graph $\mathcal{G}$ associated to $\mathcal{C}$ if for all $i \neq j, i \in \mathcal{I}, j \in V$, $D\left(P_{X_j}^{\mathcal{C}, (\emptyset)}, P_{X_j}^{\mathcal{C}, do(X_i := \tilde{N}_i)}\right) > \epsilon$ if and only if there is a directed path from $i$ to $j$ in $\mathcal{G}$.

The concept of interventional faithfulness, where interventions on a node cause significant deviations in the distribution of downstream nodes, is discussed in various contexts. It aligns closely with the notion of "c-faithfulness" in causal graphs, where interventional distributions respect the causal structure and result in significant changes in downstream nodes (Shanmugam et al., 2015; Squires et al., 2020), and with the notion of "influentiality" (Tian & Pearl (2001), Definition 2). Assumption 4.4 in Yang et al. (2018) explicitly states that interventions on upstream nodes should affect downstream nodes. The causal Markov condition and faithfulness for both observational and interventional distributions also support this idea (Addanki & Kasiviswanathan, 2021). The use of interventional data to improve the identifiability of causal models is another related concept, emphasizing how interventions can reveal causal relationships not identifiable from observational data alone (Hauser & Bühlmann, 2014).

**Lemma 2.** *If all interventions in $\tilde{N}$ have full support and all directed paths in $\mathcal{G}$ imply a total causal effect, then $(\tilde{N}, \mathcal{C})$ is $\epsilon$-interventionally faithful for $\epsilon = 0$. (Follows from Proposition 6.13 in Peters et al. (2017).)*

**Lemma 3.** *Consider a linear structural causal model $\mathcal{C}$ defined by the equations $X_j = \sum_{i \in Pa_j} \beta_{ij} X_i + N_j$, where $\beta_{ij}$ are the weights representing causal effects, drawn independently from a continuous distribution, and $N_j$ are noise variables that are independent of each other and of the parent variables, each with a distribution having full support. Let the $\tilde{N}_i$ be distributions with full support. Then, $(\tilde{N}, \mathcal{C})$ is almost surely $\epsilon$-interventionally faithful for $\epsilon = 0$, as the total causal effect along any directed path is non-zero with probability one.*

*Remark.* If $(\tilde{N}, \mathcal{C})$ is $\epsilon$-interventionally faithful for $\epsilon = 0$, then it is also $\epsilon$-interventionally faithful for all $0 < \epsilon < \min\{D\left(P_{X_j}^{\mathcal{C},(\emptyset)}, P_{X_j}^{\mathcal{C},do(X_i := \tilde{N}_i)}\right), \forall(i, j) \in E : D\left(P_{X_j}^{\mathcal{C},(\emptyset)}, P_{X_j}^{\mathcal{C},do(X_i := \tilde{N}_i)}\right) > 0\}$

Lemma 3 shows that the $\epsilon$-interventionally faithful assumption covers a large class of SCMs. We leave further theoretical work to characterize how large the class of $\epsilon$-interventionally faithful $(\tilde{N}, \mathcal{C})$ tuples is as future work.

## 5 LATENT CONFOUNDERS

Until now, we have assumed that all causal variables are observed, an assumption called *causal sufficiency*. We here discuss how the notion of interventional faithfulness and the results of this paper are applicable to the case where only a subset of the causal variables are observed. This is a common situation in practice either because we want to focus on a subset of variables which are of our interest, or because we are certain about the existence of unobserved confounders. To that end, we take the common formalization of marginalized SCMs, that marginalizes the latent confounders from the joint distribution and gives rise to a new SCM where the effects of marginalized variables are now reflected in functional dependencies among observed variables. Therefore, some variables become confounded as they depend on the same noise variables. An important property of acyclic SCMs is that they are closed under marginalizations and that their marginalization respects the latent projection (Bongers et al., 2021). The latent projection describes how to construct a new graph among the remaining variables, representing the appearance of confounders, or dependent noise variables, with bidirected (hyper) edges (Verma, 1991; Evans, 2016; 2018b). Those two representations are referred to as Acyclic Directed Mixed Graphs (ADMGs) (Verma, 1991; Andrews et al., 2022; Verma & Pearl, 2022; Richardson, 2003) and mDAGs (Evans, 2016; 2018a; Bongers et al., 2021). For our purpose, both representations are equivalent, and we thus focus on ADMGs. For the DAG $\mathcal{G}$ of the original SCM, we write the projection over a subset of variables $V' \subset V$ as $\mathcal{G}' = (V', E', B)$, where $B$ is a set of bidirected edges, and the marginalized SCM as $\mathcal{C}'$.

**Proposition 1.** *If the tuple $(\tilde{N}, \mathcal{C})$ is $\epsilon$-interventionally faithful to the graph $\mathcal{G}$ associated to $\mathcal{C}$, then for any marginalization $V' \subset V$, the marginalized SCM $\mathcal{C}'$ and the subset of interventional distributions $\tilde{N}' = \bigcup_{k \in \mathcal{I} \cap V'} \{\tilde{N}_k\}$ is $\epsilon$-interventionally faithful to $(V', E')$ from the projected ADMG $\mathcal{G}' = (V', E', B)$.*

This result derives directly from the fact that ancestral relationships are preserved in the projected graph, and that the (interventional) distributions induced by the marginalized SCM correspond to the marginals of the distributions induced by the original SCM (Bongers et al., 2021). As such, our algorithm is applicable to settings with confounders or on subsets of variables, and the theoretical guarantees hold for the projected DAG $(V', E')$. However, we note that we here implicitly assume that there is no selection bias in the sampling process (e.g., preferential inclusion of certain data points or experiments), and that latent confounders remain stable across conditions.

## 6 A SCORE ON CAUSAL ORDERS

Given an observational distribution $P_X^{\mathcal{C},(\emptyset)}$ and a set of interventional distributions $\mathcal{P}_{int} = \{P_X^{\mathcal{C},do(X_k := \tilde{N}_k)}, k \in \mathcal{I}\}$, $\mathcal{I} \subseteq V$, we define the following score for a permutation $\pi$, for some statistical distance $D : \mathcal{P}(M) \times \mathcal{P}(M) \to [0, \infty)$, $\epsilon > 0$, $c > \epsilon$:

$$S(\pi, \epsilon, D, \mathcal{I}, P_X^{\mathcal{C},(\emptyset)}, \mathcal{P}_{int}, c) = \sum_{\pi(i) < \pi(j), i \in \mathcal{I}, j \in V} \left( D\left(P_{X_j}^{\mathcal{C},(\emptyset)}, P_{X_j}^{\mathcal{C},do(X_i := \tilde{N}_i)}\right) - \epsilon\right)$$
$$+ c \cdot d \cdot \mathbf{1}_{D\left(P_{X_j}^{\mathcal{C},(\emptyset)}, P_{X_j}^{\mathcal{C},do(X_i := \tilde{N}_i)}\right) > \epsilon} \tag{1}$$

Intuitively, the sum quantifies how the causal order corresponds to strong causal effects. The rescaling from the second term by a factor of $d$ ensures that effects higher than $\epsilon$ will force the optimal solution to have $\pi(i) < \pi(j)$ by inflating their weight compared to effects smaller than $\epsilon$. Usually, we can set the $\epsilon$ of eq. (1) to the same value of $\epsilon$-interventional faithfulness as the input dataset, unless it is equal to zero. However, the remark in section 4 ensures the existence of a suitable $\epsilon$ in that case, as it can be chosen in a data-driven way by taking a non-zero value for $\epsilon$ smaller than the smallest non-negative distance.

**Theorem 1.** *Assume that we are given $P_X^{\mathcal{C},(\emptyset)}$ and $P_X^{\mathcal{C},do(X_k := \tilde{N}_k)}, \forall k \in \mathcal{I}$ and $\mathcal{I} = V$, such that $(\tilde{N}, \mathcal{C})$ is $\epsilon$-interventionally faithful for some $\epsilon > 0$, and let $\pi_{opt} \in \arg\max_\pi \mathcal{S}(\pi)$. Then $D_{top}(\mathcal{G}, \pi_{opt}) = D_{top}(\mathcal{G}, \pi^*) = 0$.*

The above theorem shows that in the case where we observe interventions on all the variables, we are guaranteed to find a valid topological order by maximizing our score when the data is $\epsilon$-interventionally faithful. Similar results were proven for this case where all the variables are intervened (Hauser & Bühlmann, 2012; Tian & Pearl, 2001). We now turn to the case where variables have a probability of being intervened smaller than one. In that case, the error will not be zero in expectation, and we thus aim to upper-bound the expected error as measured by $D_{top}$. We first prove an important property of the optimum to our proposed score. Lemma 4 shows that for a given edge, a single intervention among a large set of candidate variables is sufficient to correctly order the two variables of an edge.

**Lemma 4.** *Assume that we are given $P_X^{\mathcal{C},(\emptyset)}$ and $P_X^{\mathcal{C},do(X_k:=\tilde{N}_k)}, \forall k \in \mathcal{I}$, such that $(\tilde{N}, \mathcal{C})$ is $\epsilon$-interventionally faithful for some $\epsilon > 0$, and let $\pi_{opt} \in \arg\max_\pi \mathcal{S}(\pi)$. Let $(i,j) \in E$, then if $j \in \mathcal{I}$ or for some $k \in \boldsymbol{AN}_j^{\mathcal{G}} \setminus \boldsymbol{AN}_i^{\mathcal{G}}, k \in \mathcal{I}$, then $\pi_{opt}(i) < \pi_{opt}(j)$.*

We now use this result to derive a graph dependent upper-bound to the expected error, given a uniform probability for a variable to be intervened. We leave theoretical derivations for other intervention distributions, for example data dependent intervention selection, as future work.

**Theorem 2.** *Assume that we are given $P_X^{\mathcal{C},(\emptyset)}$ and $P_X^{\mathcal{C},do(X_k:=\tilde{N}_k)}, \forall k \in \mathcal{I}$, such that $(\tilde{N}, \mathcal{C})$ is $\epsilon$-interventionally faithful for some $\epsilon > 0$, and let $\mathcal{I}$ be chosen uniformly at random, where $p_{int} := P(i \in \mathcal{I})\forall i \in V, 0 < p_{int} < 1$, then $\mathbb{E}[D_{top}(\mathcal{G}, \pi_{opt})] \leq \sum_{(i,j)\in\mathcal{G}}(1-p_{int})^{|\boldsymbol{AN}_j^{\mathcal{G}}\cup\{j\}\setminus\boldsymbol{AN}_i^{\mathcal{G}}|}$.*

# 7 RELAXATION OF $\epsilon$-INTERVENTIONAL FAITHFULNESS

We now look at the case where $\epsilon$-interventional faithfulness may not be fully fulfilled and we derive some bounds for that setting. We consider the worst case to be the one where interventions only reveal direct causal relationships, i.e., only children are affected by a node intervention such that the distance is larger than $\epsilon$. We can hypothesize that real distributions lie on a continuum between those two extremes, that is, some interventions have an $\epsilon$-strong effect only on children and other interventions have an effect on more downstream variables. This restricted setting also covers the case when multiple paths between two variables $i$ and $j$ "cancel out" the causal effect of $i$ on $j$. In that restricted setting, we can rewrite lemma 4 as follows:

**Lemma 5.** *Let $(i,j) \in E$, then if $j \in \mathcal{I}$ or for some $k \in \boldsymbol{Pa}_j^{\mathcal{G}} \setminus \boldsymbol{Pa}_i^{\mathcal{G}}, k \in \mathcal{I}$, then $\pi_{opt}(i) < \pi_{opt}(j)$.*

It is then straightforward that the bound on $D_{top}$ for $\pi_{opt}$ in that case is:

**Theorem 3.** *Let $\mathcal{I}$ be chosen uniformly at random, where $p_{int} := P(i \in \mathcal{I})\forall i \in V, 0 < p_{int} < 1$, then $\mathbb{E}[D_{top}(\mathcal{G}, \pi_{opt})] \leq \sum_{(i,j)\in\mathcal{G}}(1-p_{int})^{|\boldsymbol{Pa}_j^{\mathcal{G}}\cup\{j\}\setminus\boldsymbol{Pa}_i^{\mathcal{G}}|}$.*

Now that the bound only depends on the parent of the variables, it becomes possible to find a closed form for the expectation of $D_{top}$ for a particular graph distribution.

**Theorem 4.** *Let $\mathcal{I}$ be chosen uniformly at random, $\mathcal{G}$ be a random Erdos-Renyi directed acyclic graph with edge probability $p_e$, where $p_{int} := P(i \in \mathcal{I})\forall i \in V, 0 < p_{int}$, then $\mathbb{E}[D_{top}(\mathcal{G}, \pi_{opt})] \leq \frac{(1-p_{int})^2}{p_{int}}\left[d - (1-p_{int}p_e)\frac{1-(1-p_{int}p_e)^d}{p_{int}p_e}\right]$. We also have a looser, but independent of $p_e$, bound $\mathbb{E}[D_{top}(\mathcal{G}, \pi_{opt})] \leq \frac{(1-p_{int})^2}{p_{int}}d$.*

The natural next step is to look at how the bound behaves as $d$ goes to infinity. More precisely, let's look at the normalized error $\frac{\mathbb{E}[D_{top}(\mathcal{G},\pi_{opt})]}{d}$. If $p_e$ is fixed, we simply get $\lim_{d\to\infty} \frac{\mathbb{E}[D_{top}(\mathcal{G},\pi_{opt})]}{d} \leq \frac{(1-p_{int})^2}{p_{int}}$. However, in realistic scenarios, we can expect the edge probability to depend on the number of variables. A common assumption is that the expected number of edges per variable is constant.

**Lemma 6.** *Let $\mathbb{E}|E| = \frac{d(d-1)}{2}p_e = \frac{c}{2}(d-1)$, where $c > 0, c \in \mathbb{R}$ is a constant, then $\lim_{d\to\infty} \frac{\mathbb{E}[D_{top}(\mathcal{G},\pi_{opt})]}{d} \leq \frac{(1-p_{int})^2}{p_{int}}\left[1 - \frac{1}{p_{int}\cdot c}(1 - e^{-p_{int}\cdot c})\right]$.*

The above lemma shows that the expected error as $d$ goes to infinity is $\mathcal{O}(d)$, providing a strong guarantee on $\pi_{opt}$ even in a large scale setting.

# 8 INTERSORT

Given that our score for causal order eq. (1) is optimized over a set of permutations, finding the optimum by exhaustively trying all possible solutions becomes intractable as the number of variables grows larger. We thus develop an algorithm to tractably find a solution that optimizes for our proposed score in eq. (1). This algorithm consists of two steps. The first step uses a different score to find an initial solution. In the second step, this solution is refined by performing a local search using our original score eq. (1).

## 8.1 INTERSORT STEP 1: FINDING AN INITIAL SOLUTION

To find an initial solution, we design a new score, similar to eq. (1), but this time on graphs. The intuition is that we want to score edges directly, and edges $(i, j)$ that have a distance $D\left(P_{X_j}^{\mathcal{C},(\emptyset)}, P_{X_j}^{\mathcal{C},do(X_i := \tilde{N}_i)}\right)$ greater than $\epsilon$ should be on the upper-triangular part. We thus write this new score $\mathcal{S}_{init}$ as follows:

$$\mathcal{S}_{init}\left(G, \epsilon, D, \mathcal{I}, P_X^{\mathcal{C},(\emptyset)}, \mathcal{P}_{int}\right) = \sum_{(i,j) \in G, i \in \mathcal{I}} D\left(P_{X_j}^{\mathcal{C},(\emptyset)}, P_{X_j}^{\mathcal{C},do(X_i := \tilde{N}_i)}\right) - \epsilon |G| \qquad (2)$$

Then, we aim to find a DAG $G$ that maximizes this score.

$$G_{opt} = \arg\max_{G \in \mathcal{G}_{DAG}} \mathcal{S}_{init}\left(G, \epsilon, D, \mathcal{I}, P_X^{\mathcal{C},(\emptyset)}, \mathcal{P}_{int}\right) \qquad (3)$$

Once again, the search over all possible DAGs is computationally expensive. We thus design an approximation algorithm based on the following heuristic: the highest values $D\left(P_{X_j}^{\mathcal{C},(\emptyset)}, P_{X_j}^{\mathcal{C},do(X_i := \tilde{N}_i)}\right)$, which can be seen as a score on the corresponding edge $(i, j)$, should be edges in the solution DAG to maximize the score $\mathcal{S}_{init}$. To do so, we sort the edges based on this "edge score" from highest to lowest. We then construct our solution by adding the edges following the sorted order, adding an edge only if it does not break the acyclity of the solution. We then stop adding edges when those have a score $D_{ij} = D\left(P_{X_j}^{\mathcal{C},(\emptyset)}, P_{X_j}^{\mathcal{C},do(X_i := \tilde{N}_i)}\right)$ lower than $\epsilon$ in the matrix $\mathbf{D} = (D_{ij})$. The topological order of this solution $G_{opt}$ for $\mathcal{S}_{init}$ is then used as initial condition for our original objective of maximizing $\mathcal{S}$. The runtime of this algorithm is dominated by the sorting step, which is $\mathcal{O}(d \cdot |\mathcal{I}| \log(d \cdot |\mathcal{I}|))$. We call this procedure SORTRANKING and illustrate it in algorithm 2 in the appendix.

## 8.2 INTERSORT STEP 2: A LOCAL SEARCH FOR $\pi_{opt}$

Starting from the initial solution found by maximizing eq. (2), we then perform a greedy local search to optimize our original objective eq. (1). At each step, given a current solution, we search in a close neighbourhood set for a candidate permutation $\pi$ with a higher score than our current solution. To derive the neighborhood set, we introduce an operator $\Sigma : \{\{1, ...d\} \to \{1, ...d\}\} \to \{\{1, ...d\} \to \{1, ...d\}\}$, defined such that for a given set of permutations $\Pi$, $\Sigma(\Pi)$ returns a new set of permutations $\Pi'$. Each permutation $\pi \in \Pi$ is modified by applying $\sigma : \{1, ...d\} \to \{1, ...d\} \to \{\{1, ...d\} \to \{1, ...d\}\}$, which for each variable $i \in V$, returns all the permutations such that $\pi(i)$ takes any of the other $d - 1$ possible values. $\sigma$ takes one permutation as input and return a set of permutations which is $\mathcal{O}(d^2)$. The total number of applications of $\Sigma$, denoted as $k$, directly defines the neighborhood size which is $\mathcal{O}(d^{2k})$. This definition allows us to explore varying depths of the search space, which is $\mathcal{O}(d!)$, by controlling $k$, thereby balancing between the breadth of the search and the computational resources required. A smaller $k$ value implies a tighter search around the initial solution and encode our confidence in the initial solution, optimizing runtime at the potential cost of missing broader, potentially better solutions. In our experiments, we use $k = 1$. We then stop once an iteration of the algorithm brings no improvement to the score. This procedure, that we name LOCALSEARCH, is described in algorithm 3 in the appendix. Our complete algorithm for finding the causal order is called INTERSORT (see algorithm 1 in the appendix).

Now arises the question of how close the solutions from the approximation algorithm INTERSORT are to the optimum of eq. (1). We evaluate this empirically for graphs with 5 variables, where computing the exact solution is still viable. The results are presented in fig. 1a. As can be observed, the mean $D_{top}$ scores of the two algorithms are very close and have overlapping confidence intervals (at level 95%).

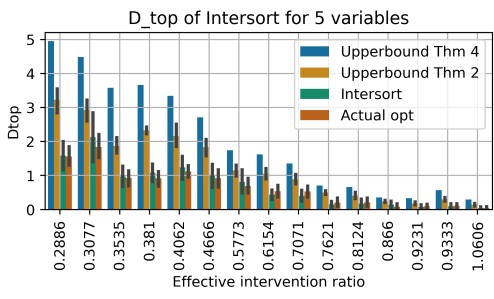 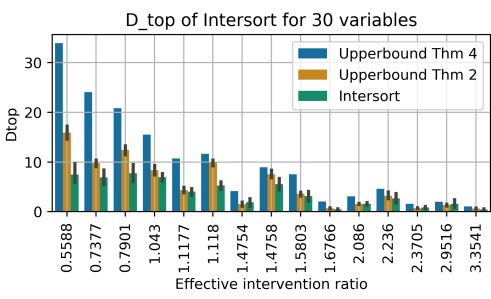

(a) Simulation with 5 variables      (b) Simulation with 30 variables

Figure 1: Simulation and comparison between the two bounds, and between Intersort and the exact $\pi_{opt}$. For each setting, we draw 20 graphs per setting, where a setting is the tuple $(p_{int}, p_e)$. Then, for each graph, we run the algorithm on 10 configurations, where each configuration corresponds to a draw of the targeted variables following $p_{int}$. For both 5 and 30 variables, we have $p_{int} \in \{0.25, 0.33, 0.5, 0.66, 0.75\}$. In the 5 variables setting on the left, we have $p_e \in \{0.5, 0.66, 0.75\}$. In the 30 variables settings on the right, we have $p_e \in \{0.05, 0.1, 0.2\}$. The settings are ordered on the x-axis following what we call the effective intervention ratio $\frac{p_{int}}{\sqrt{p_e}}$. We observe that the error is approximately monotonic when ordered by the effective intervention ratio.

We note that more refined algorithms could be developed to solve this optimization over permutations. For example, for large $d$, the local search may become computationally expensive. However, for the purpose of this study, we find the proposed algorithm to have satisfying properties, as it scales to settings with 30 variables and finds a solution close to the optimum. We also report results for a scale-free network modelled with the Barabasi-Albert distribution Albert & Barabási (2002) in the appendix in fig. 3, which supports the generality of our approach. We also report the performance of SORTRANKING in large scale settings in figs. 4 and 5. The results demonstrate that the our score can be optimized at scale, even though we can observe room for improvement, as the error is above the upper-bounds for many settings. As such, we leave further algorithmic development for large scale settings as potential future work.

## 9   ESTIMATING THE DISTANCES FROM I.I.D SAMPLES

We now turn to the question of computing the statistical distances between observational and interventional distributions from real samples. Indeed, until now, we assumed access to population distributions, such that $D(P, Q) = 0$ if and only if $P = Q$. However, in a real world setting, we may only assume that we have access to $i.i.d$ samples from the distributions, that is $\{X^i\}_{i=1}^{n_0} \sim P_X^{\mathcal{C},(\emptyset)}$ and for $j \in \{1, \ldots, d\}, \{X^i\}_{i=1}^{n_j} \sim P_X^{\mathcal{C}, do(X_j := \tilde{N}_j)}$, where $n_0, \ldots, n_d \in \mathbb{N}$ are the sample sizes of the observational set and of the interventional sets. Given this setting, we require a statistical distance with the following properties: it should be applicable to sample distributions, the distance between $i.i.d$ samples should converge to the distance between the population distribution as $n$ goes to infinity, with some guarantees on the convergence rate. A distance that fulfills all those criteria is the Wasserstein distance Villani et al. (2009). More specifically, it converges to the population distance in $\mathcal{O}\left(\frac{\sqrt{\ln n}}{\sqrt{n}}\right)$, where $n$ is the number of samples. We recall the definition of this distance in the appendix.

## 10   EMPIRICAL RESULTS

We now evaluate Intersort on simulated empirical data and compare its performance to various baselines. The graphs are simulated from a Erdos-Renyi distribution (Erdős et al., 1960), with an expected number of edges per variable $c \in \{1, 2\}$. We follow a setup close to (Lorch et al., 2022) to simulate data from linear and random Fourier features (RFF) additive functional relationships. We also apply the models to simulated single-cell data from the SERGIO (Dibaeinia & Sinha, 2020) model, using the code from Lorch et al. (2022) (MIT License, v1.0.5). Lastly, we apply on neural network functional data following the setup of Brouillard et al. (2020) and using the implementation of Nazaret et al. (2023) (MIT License, v0.1.0). We run the models on varying

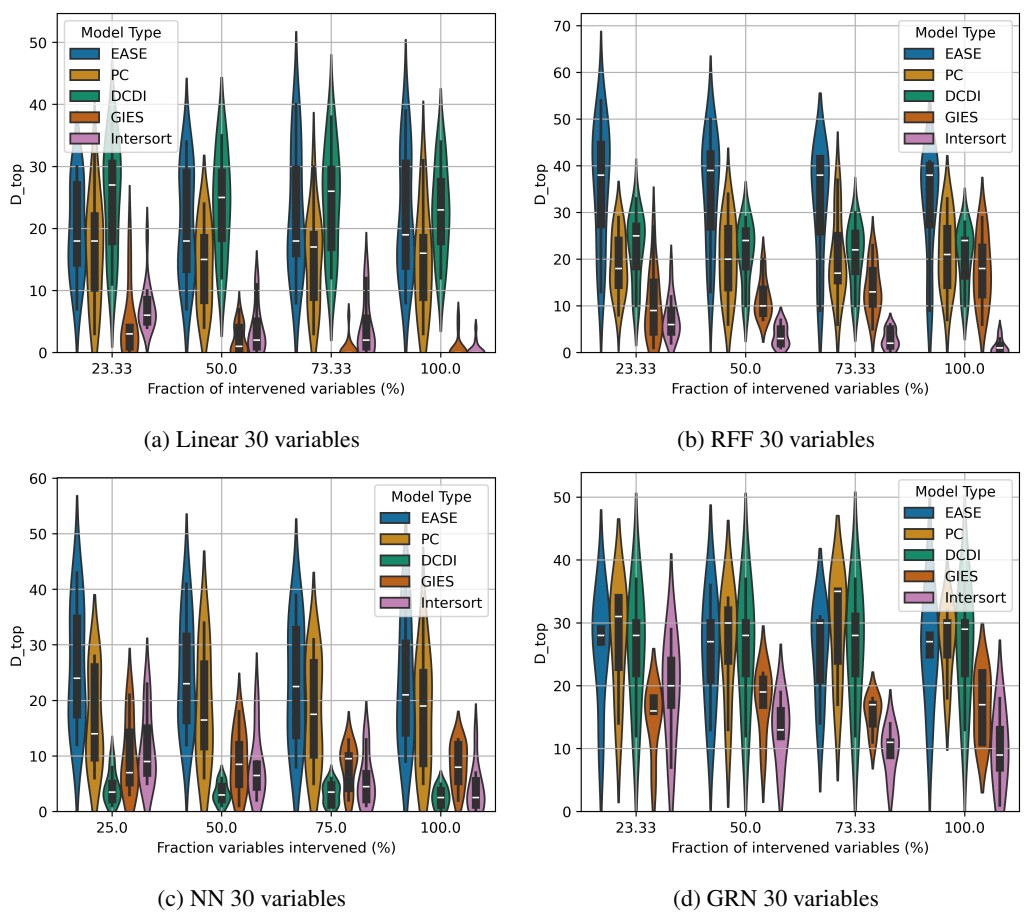

Figure 2: Comparison of the performance of the baselines and of our model INTERSORT across diverse data domains as presented (linear, RFF, NN and GRN data), for 30 variables. The x-axis corresponds to the fraction of variables that have been targeted by an intervention. The y-axis is the performance of causal ordering prediction as measured by the $D_{top}$ metric (see section 3, lower is better). The violins are order from left to right: EASE, PC, DCDI, GIES, Intersort. Results for 10 and 100 variables can be found in the appendix (figs. 7 and 8).

levels of intervention, where the ratio of intervened variables is in $\{25\%, 50\%, 75\%, 100\%\}$. The data is always standardized based on the mean and variance of the observational dataset, which removes the Varsortability (Reisach et al., 2021) artefact in the data. For the RFF and linear domain, the distribution of the noise is chosen uniformly at random from uniform Gaussian (noise scale independent from the parents), heteroscedastic Gaussian (noise scale functionally dependent on the parents), and Laplace. For the neural network domain, the noise distribution is Gaussian with a fixed variance. We run on 10 simulated datasets for each domain and each ratio of intervened variables. We simulate 5000 samples for the observational datasets and 100 samples for each interventions, which is a setting similar to real single cell transcriptomics datasets (Replogle et al., 2022).

Choosing appropriate baselines to compare to is not trivial, given that, to our knowledge, our setting of predicting the causal order from interventional data has not been considered in the literature, as the focus in prior literature has been on observational data (Reisach et al., 2021; Bühlmann et al., 2014; Rolland et al., 2022; Shimizu et al., 2011). As such, we construct causal orders from the CPDAG predicted by two standard methods, namely PC (Spirtes et al., 2000) and GIES (Hauser & Bühlmann, 2012), by creating a DAG from the oriented edges and then computing the topological order of this DAG. PC is a constrained based method relying on conditional independence tests to estimate the structure of the graph. We run the implementation of Zheng et al. (2024) (MIT License, causal-learn v0.1.3). GIES is an extension to interventional data of the GES model (Chickering, 2002), which is a score based method that greedily adds and removes edges to the estimated CPDAG. We use

the Gaussian BIC score, and run the package implementation of Gamella (2022) (BSD 3-Clause License, v0.0.1). Additionally, we compare to EASE (Gnecco et al., 2021), which is a method that aims to learn the causal order from observational data, leveraging the insight that extreme tail-values reveal the causal order. Given that interventions can lead to extreme values not present in the observational data, we also compare to this method. For EASE, we run our own Python implementation of the algorithm. Lastly, we compare to DCDI (Brouillard et al., 2020), which is a continuously differentiable causal discovery model that can leverage interventional data. For Intersort, we use $\epsilon = 0.3$ for the linear, RFF and neural network domain, and $\epsilon = 0.5$ for the single-cell domain. We set $c = 0.5$. We compute the Wassertein distances with the SCIPY python package (Virtanen et al., 2020).

The results for each method on the different types of data considered are displayed in fig. 2, where the distribution of the $D_{top}$ score of each model is plotted against the ratios of intervened variables considered. As can be observed, INTERSORT performs well on all data domains, and shows decreasing error as more interventions are available, exhibiting the model's capability to capitalize on the interventional information to recover the causal order across diverse settings. Compared to the baselines, only GIES in the linear domain and DCDI in the neural network domain at $25\%$ and $50\%$ perform better. These experiments demonstrate that $\epsilon$-interventional faithfulness is fulfilled by a diverse set of data types, and that this property can be robustly exploited to recover causal information. Only when strong assumptions about the data distributions are fulfilled, such as linear data for GIES, can better results be obtained in low intervention settings.

## 11    DISCUSSION AND CONCLUSION

We note that INTERSORT's performance could still be improved in various ways. For example, a more scalable or closer to the optimum approximation algorithm could be designed. Scalability of Intersort in its current form is mainly constrained by the LOCALSEARCH component, for which alternative optimization techniques could be employed to enhance efficiency and manage larger variable sets. Also, the optimal parameter $\epsilon$ could be chosen in a data dependent way instead of using a fixed value. Lastly, other statistical distances, especially ones that converge faster to $0$ in terms of number of samples when the two distributions are equal, could lead to better results. We leave such potential improvements as future work. Another question that may need further investigation is the $\epsilon$-interventional faithfulness assumption. We argue that it is not any stronger than common assumption in the literature such as faithfulness, and lemma 3 shows that it already covers a sizeable set of SCMs. Furthermore, our empirical results demonstrate that it probably holds for a rather large class of SCMs. We also derive theoretical results for a relaxation of this assumption in section 7. An interesting direction of theoretical work would be to further analyze the intricate relationship between the statistical distance $D$, the number of variables $d$, the parameter $\epsilon$, the interventional random variables $\tilde{N}$, and the sample size for non-asymptotic settings. For example, our framework makes it amenable to study the strength of the interventions needed to reveal the causal descendants above a statistically significant threshold.

In this study, we introduced INTERSORT, an innovative algorithm designed to uncover the causal order from interventional data by optimizing for a new proposed score on causal orders. We introduced the $\epsilon$-interventional faithfulness assumption and proved that interventional datasets fulfilling it have strong guarantees in terms of upper-bounds on the expected error of the optimum of our score. The performance of INTERSORT and the validity of our theoretical findings was extensively demonstrated on a diverse set of simulated datasets, across functional relationships, noise types and graph densities. INTERSORT has demonstrated superior performance when compared to established benchmarks such as PC, GIES, DCDI and EASE. The robustness and versatility of INTERSORT underscore the potential of the $\epsilon$-interventional faithfulness assumption to reshape causal inference methodologies. We envision that this work should spearhead new model development based on our proposed perspective on dataset with high numbers of single variable interventions. This includes downstream tasks such as causal discovery, for example by reconstructing the graph given the estimated causal order. It could inform active intervention selection as well, as designing a policy to choose which variables to target next may lead to better guarantees than our results based on interventions selected uniformly at random. All these directions and method developments can have real world impact on critical application domains such as biology, from which this work and its assumptions were inspired.

## ACKNOWLEDGMENTS AND DISCLOSURE

The authors thank Djordje Miladinovic, Aleksei Triastcyn and Lachlan Stuart for feedback and edits on the manuscript. The authors also thank Prof. Nicolai Meinshausen for valuable suggestions on the theoretical part, as well as Prof. Stefan Bauer for discussions about related work and relevance of the project. MC, AM and PS are employees and shareholders of GSK plc.

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

## A  PROOFS OF THEOREMS

*Proof of Lemma 3.* Recall that for a linear SCM, the total causal effect between any pair $(i,j)$ is equal to the sum of the products of the weights along each paths from $i$ to $j$. Given that the weights are continuously random and independent, the probability that the sum is exactly 0 is 0. As such, the tuple $(\tilde{N}, \mathcal{C})$ is $\epsilon$-interventionally faithful for $\epsilon = 0$ almost surely. $\square$

*Proof of Theorem 1.* Let $\mathbf{D}$ be a $d \times d$ matrix containing all the distances, where $\mathbf{D}(i,j) = D\left(P_{X_j}^{\mathcal{C},(\emptyset)}, P_{X_j}^{\mathcal{C},do(X_i := \tilde{N}_i)}\right)$. For the case of every variables being intervened on, we can set $c = 0$.

We are going to prove that $\overline{A}_{\pi_{opt}}$ is upper-triangular, and from that it follows that $D_{top}(\mathcal{G}, \pi_{opt}) = 0$. We know that such a permutation exists, as for any $\pi^*$, $\overline{A}_{\pi^*}$ is upper-triangular.

Second, we note that given that $(\tilde{N}, \mathcal{C})$ is interventionally faithful, $\overline{A}$ has the non zero entries at the same position than the matrix $D$. We thus need to show that for any $D_\pi$ that is not upper-triangular, then $\pi \notin \arg\max_\pi \mathcal{S}(\pi)$. Then we can compute:

$$
\begin{aligned}
\mathcal{S}(\pi^*) &= \sum_{\pi^*(i) < \pi^*(j),} (\mathbf{D}(i,j) - \epsilon) \\
&= \sum_{\pi^*(i) < \pi^*(j)} \mathbf{D}(i,j) - \frac{d \cdot (d-1)}{2} \epsilon \\
&= \sum_{i,j} \mathbf{D}(i,j) - \frac{d \cdot (d-1)}{2} \epsilon
\end{aligned}
$$

Then, let $\pi$ be such that $D_\pi$ has some non-zero entries in the lower-triangular part. Then:

$$
\mathcal{S}(\pi^*) - \mathcal{S}(\pi) = \sum_{\pi(i) > \pi(j)} \mathbf{D}(i,j) > 0,
$$

which implies that $\pi \notin \arg\max_\pi \mathcal{S}(\pi)$. $\square$

*Proof of lemma 4.* Let $k$ be a variable that satisfies the condition, that is, $k \in \mathcal{I}$ and $k \in \mathbf{AN}_j^{\mathcal{G}} \setminus \mathbf{AN}_i^{\mathcal{G}}$. Then, for this $k$, the hypothesis implies that $D\left(P_{X_j}^{\mathcal{C},(\emptyset)}, P_{X_j}^{\mathcal{C},do(X_k := \tilde{N}_k)}\right) > \epsilon$ and $D\left(P_{X_i}^{\mathcal{C},(\emptyset)}, P_{X_i}^{\mathcal{C},do(X_k := \tilde{N}_k)}\right) = 0$. We thus need to show that given those distances, $\pi_{opt}$ is such that $\pi_{opt}(i) < \pi_{opt}(j)$.

Let's note that only distances that are larger than $\epsilon$ contribute positively to the score. All the distances that are equal to 0 contribute $-\epsilon$ to the sum. As such, a necessary condition for an optimal permutation is that all distances $D\left(P_{X_i}^{\mathcal{C},(\emptyset)}, P_{X_i}^{\mathcal{C},do(X_j := \tilde{N}_k)}\right) > \epsilon$ contribute to the sum, which happens only if $\pi_{opt}(i) < \pi_{opt}(j)$. Given that $(\tilde{N}, \mathcal{C})$ is $\epsilon$-interventionally faithful, we have that any $\pi \in \Pi^*$ are such that all distances larger than $\epsilon$ contribute to $\mathcal{S}(\pi)$, which shows existence of such a permutation. $\pi_{opt}$ thus must have that $\pi_{opt}(k) < \pi_{opt}(j)$. If $i \in \mathcal{I}$, then with the same argument we have $\pi_{opt}(i) < \pi_{opt}(j)$.

It remains to show that $\pi_{opt}(i) < \pi_{opt}(k)$ if $i \notin \mathcal{I}$. We have $D\left(P_{X_i}^{\mathcal{C},(\emptyset)}, P_{X_i}^{\mathcal{C},do(X_k := \tilde{N}_k)}\right) = 0$, which contributes negatively to the score if $\pi_{opt}(k) < \pi_{opt}(i)$. So changing to $\pi_{opt}(i) < \pi_{opt}(k)$ improves the score by $\epsilon$, as $i \notin \mathcal{I}$. This implies that $\pi_{opt}(i)$ must be smaller than $\pi_{opt}(k)$.

Lastly, if $j \in \mathcal{I}$, but not $i$ and no $k \in \mathbf{AN}_j^{\mathcal{G}} \setminus \mathbf{AN}_i^{\mathcal{G}}$, we need to show that we still have $\pi_{opt}(i) < \pi_{opt}(j)$. With the same argument as before, having $\pi_{opt}(i) < \pi_{opt}(j)$ can only improves the score as $i \notin \mathcal{I}$. This concludes the overall proof. $\square$

*Proof of theorem 2.*

$$\mathbb{E}[D_{top}(\mathcal{G}, \pi_{opt})] = \mathbb{E} \sum_{(i,j) \in \mathcal{G}} 1_{\pi_{opt}(i) > \pi_{opt}(j)}$$

$$= \sum_{(i,j) \in \mathcal{G}} \mathbb{E}[1_{\pi_{opt}(i) > \pi_{opt}(j)}]$$

$$= \sum_{(i,j) \in \mathcal{G}} P(\pi_{opt}(i) > \pi_{opt}(j))$$

Now let's look at $P(\pi_{opt}(i) > \pi_{opt}(j))$, where there is an edge from $i$ to $j$ in the graph. Let's note the condition of lemma 4 as $C$.

$$P(\pi_{opt}(i) > \pi_{opt}(j)) = P(\pi_{opt}(i) > \pi_{opt}(j)|\neg C)P(\neg C) + P(\pi_{opt}(i) > \pi_{opt}(j)|C)P(C)$$

$$= P(\pi_{opt}(i) > \pi_{opt}(j)|\neg C)P(\neg C)$$

$$= P(\pi_{opt}(i) > \pi_{opt}(j)|\neg C)P(\forall k \in \mathbf{AN}_j^{\mathcal{G}} \cup \{j\} \setminus \mathbf{AN}_i^{\mathcal{G}} : k \notin \mathcal{I})$$

$$= P(\pi_{opt}(i) > \pi_{opt}(j)|\neg C)(1 - p_{int})^{|\mathbf{AN}_j^{\mathcal{G}} \cup \{j\} \setminus \mathbf{AN}_i^{\mathcal{G}}|}$$

$$\leq (1 - p_{int})^{|\mathbf{AN}_j^{\mathcal{G}} \cup \{j\} \setminus \mathbf{AN}_i^{\mathcal{G}}|}$$

We can then take the sum over all the edges to conclude the proof.

$\square$

*Proof of theorem 4.* Without loss of generality, we assume that the variable are already in the causal order. We then write the error as a sum on all the upper-triangular elements that contain an edge and for which the variable are misplaced by $\pi_{opt}$.

$$\mathbb{E}[D_{top}(\mathcal{G}, \pi_{opt})] = \mathbb{E} \sum_{i=1}^{d-1} \sum_{j=i+1}^{d} 1_{(i,j) \in E, \pi_{opt}(i) > \pi_{opt}(j)}$$

$$= \sum_{i=1}^{d-1} \sum_{j=i+1}^{d} P((i,j) \in E, \pi_{opt}(i) > \pi_{opt}(j))$$

$$= \sum_{i=1}^{d-1} \sum_{j=i+1}^{d} p_e \cdot P(\pi_{opt}(i) > \pi_{opt}(j)|(i,j) \in E)$$

We now upper-bound the probability of error given the existence of an edge:

$$P(\pi_{opt}(i) > \pi_{opt}(j)|(i,j) \in E) \leq P(j \notin \mathcal{I} \wedge \forall k < j, k \in \mathbf{Pa}_j^{\mathcal{G}} \wedge k \notin \mathbf{Pa}_i^{\mathcal{G}} \implies k \notin \mathcal{I})$$

$$= (1 - p_{int}) \prod_{k=1}^{i} P(k \in \mathbf{Pa}_j^{\mathcal{G}} \wedge k \notin \mathbf{Pa}_i^{\mathcal{G}} \implies k \notin \mathcal{I}) \prod_{k=i+1}^{j-1} P(k \in \mathbf{Pa}_j^{\mathcal{G}} \implies k \notin \mathcal{I})$$

$$\leq (1 - p_{int})^2 \prod_{k=i+1}^{j-1} P(k \in \mathbf{Pa}_j^{\mathcal{G}} \implies k \notin \mathcal{I})$$

$$= (1 - p_{int})^2 \prod_{k=i+1}^{j-1} (1 - P(k \in \mathbf{Pa}_j^{\mathcal{G}} \wedge k \in \mathcal{I}))$$

$$= (1 - p_{int})^2 (1 - p_{int} p_e)^{j-i-1}$$

Plugging this back into the original sum and working out the geometric sums:

$$\mathbb{E}[D_{top}(\mathcal{G}, \pi_{opt})] \le p_e(1 - p_{int})^2 \sum_{i=1}^{d-1} \sum_{j=i+1}^{d} (1 - p_{int}p_e)^{j-i-1}$$

$$= p_e(1 - p_{int})^2 \sum_{i=1}^{d-1} \sum_{j=0}^{d-i-1} (1 - p_{int}p_e)^{j}$$

$$= p_e(1 - p_{int})^2 \sum_{i=1}^{d-1} \frac{1 - (1 - p_{int}p_e)^{d-i}}{p_{int}p_e}$$

$$= \frac{p_e(1 - p_{int})^2}{p_{int}p_e} \left[ d - \sum_{i=0}^{d-1} (1 - p_{int}p_e)^{d-i} \right]$$

$$= \frac{(1 - p_{int})^2}{p_{int}} \left[ d - \sum_{i=1}^{d} (1 - p_{int}p_e)^{i} \right]$$

$$= \frac{(1 - p_{int})^2}{p_{int}} \left[ d - (1 - p_{int}p_e) \frac{1 - (1 - p_{int}p_e)^{d}}{p_{int}p_e} \right]$$

$$= \frac{(1 - p_{int})^2}{p_{int}^2 p_e} \left[ p_{int}p_e \cdot d - (1 - p_{int}p_e) + (1 - p_{int}p_e)^{d+1} \right]$$

$$\le \frac{(1 - p_{int})^2}{p_{int}} d$$

$\square$

*Proof of lemma 6.* First, we can rewrite the equation on $p_e$ to get $p_e = \frac{c}{d}$. Then, plugging it into the bound we get:

$$\mathbb{E}[D_{top}(\mathcal{G}, \pi_{opt})] \le \frac{(1 - p_{int})^2}{p_{int}} \left[ d - (1 - p_{int}p_e) \frac{1 - (1 - p_{int}p_e)^{d}}{p_{int}p_e} \right]$$

$$= \frac{(1 - p_{int})^2}{p_{int}} \left[ d - (1 - p_{int}\frac{c}{d}) \frac{d}{c} \frac{1 - (1 - p_{int}\frac{c}{d})^{d}}{p_{int}} \right]$$

Finally, we normalize by $d$ and take the limit:

$$\lim_{d \to \infty} \frac{\mathbb{E}[D_{top}(\mathcal{G}, \pi_{opt})]}{d} \le \lim_{d \to \infty} \frac{(1 - p_{int})^2}{p_{int}} \left[ 1 - (1 - p_{int}\frac{c}{d}) \frac{1 - (1 - p_{int}\frac{c}{d})^{d}}{c \cdot p_{int}} \right]$$

$$= \frac{(1 - p_{int})^2}{p_{int}} \left[ 1 - \frac{1 - e^{-c \cdot p_{int}}}{c \cdot p_{int}} \right]$$

$\square$

## B  PEN AND PAPER EXAMPLE

We here spell out all the possible configurations for a graph with 3 variables with the associated expected error of $\pi_{opt}$ in table 1 and appendix B. Readers may find those example useful to build intuition on how the algorithm works.

Table 1: Expected top-order divergence for graphs of size 3, with 2 out of the three variables observed under intervention.

| Number edges | Adjacency Matrix | Transitive Closure | $\mathbb{E}[D_{top}(\mathcal{G}, \pi_{opt})]$ |
|---|---|---|---|
| 0 | $\begin{pmatrix} 0 & 0 & 0 \\ 0 & 0 & 0 \\ 0 & 0 & 0 \end{pmatrix}$ | $\begin{pmatrix} 0 & 0 & 0 \\ 0 & 0 & 0 \\ 0 & 0 & 0 \end{pmatrix}$ | 0 |
| 1 | $\begin{pmatrix} 0 & 1 & 0 \\ 0 & 0 & 0 \\ 0 & 0 & 0 \end{pmatrix}$ | $\begin{pmatrix} 0 & 1 & 0 \\ 0 & 0 & 0 \\ 0 & 0 & 0 \end{pmatrix}$ | 0 |
| 2 | $\begin{pmatrix} 0 & 1 & 0 \\ 0 & 0 & 1 \\ 0 & 0 & 0 \end{pmatrix}$ | $\begin{pmatrix} 0 & 1 & 1 \\ 0 & 0 & 1 \\ 0 & 0 & 0 \end{pmatrix}$ | 0 |
| 2 | $\begin{pmatrix} 0 & 1 & 1 \\ 0 & 0 & 0 \\ 0 & 0 & 0 \end{pmatrix}$ | $\begin{pmatrix} 0 & 1 & 1 \\ 0 & 0 & 0 \\ 0 & 0 & 0 \end{pmatrix}$ | 0 |
| 3 | $\begin{pmatrix} 0 & 1 & 1 \\ 0 & 0 & 1 \\ 0 & 0 & 0 \end{pmatrix}$ | $\begin{pmatrix} 0 & 1 & 1 \\ 0 & 0 & 1 \\ 0 & 0 & 0 \end{pmatrix}$ | 0 |

Table 2: Expected top-order divergence for graphs of size 3, with 1 out of the three variables observed under intervention.

| Number edges | Adjacency Matrix | Transitive Closure | $\mathbb{E}[D_{top}(\mathcal{G}, \pi_{opt})]$ |
|---|---|---|---|
| 0 | $\begin{pmatrix} 0 & 0 & 0 \\ 0 & 0 & 0 \\ 0 & 0 & 0 \end{pmatrix}$ | $\begin{pmatrix} 0 & 0 & 0 \\ 0 & 0 & 0 \\ 0 & 0 & 0 \end{pmatrix}$ | 0 |
| 1 | $\begin{pmatrix} 0 & 1 & 0 \\ 0 & 0 & 0 \\ 0 & 0 & 0 \end{pmatrix}$ | $\begin{pmatrix} 0 & 1 & 0 \\ 0 & 0 & 0 \\ 0 & 0 & 0 \end{pmatrix}$ | $\frac{1}{3} \cdot \frac{1}{2} = \frac{1}{6}$ |
| 2 | $\begin{pmatrix} 0 & 1 & 0 \\ 0 & 0 & 1 \\ 0 & 0 & 0 \end{pmatrix}$ | $\begin{pmatrix} 0 & 1 & 1 \\ 0 & 0 & 1 \\ 0 & 0 & 0 \end{pmatrix}$ | $\frac{1}{3} \cdot \frac{1}{2} + \frac{1}{3} \cdot \frac{1}{2} = \frac{1}{3}$ |
| 2 | $\begin{pmatrix} 0 & 1 & 1 \\ 0 & 0 & 0 \\ 0 & 0 & 0 \end{pmatrix}$ | $\begin{pmatrix} 0 & 1 & 1 \\ 0 & 0 & 0 \\ 0 & 0 & 0 \end{pmatrix}$ | $\frac{1}{3} \cdot \frac{1}{2} + \frac{1}{3} \cdot \frac{1}{2} = \frac{1}{3}$ |
| 3 | $\begin{pmatrix} 0 & 1 & 1 \\ 0 & 0 & 1 \\ 0 & 0 & 0 \end{pmatrix}$ | $\begin{pmatrix} 0 & 1 & 1 \\ 0 & 0 & 1 \\ 0 & 0 & 0 \end{pmatrix}$ | $\frac{1}{3} \cdot \frac{1}{2} + \frac{1}{3} \cdot \frac{1}{2} = \frac{1}{3}$ |

## C    PSEUDOCODES

---

**Algorithm 1** Complete algorithm

---

1: **procedure** INTERSORT( $\mathcal{S}$: score function eq. (1), $\epsilon$,
   $D$: statistical distance function,
   $\mathcal{I}$: index set of intervened variables,
   $P_X^{\mathcal{C},(\emptyset)}$: observational distribution,
   $\mathcal{P}_{int}$: interventional distributions )
2:     Initialize $\mathbf{D}$ as a zero matrix of size $d \times d$
3:     **for** each $(i,j) \in [1..d]$ **do**
4:         **if** $i \in \mathcal{I}$ **then**
5:             $\mathbf{D}[i,j] \leftarrow D\left(P_{X_j}^{\mathcal{C},(\emptyset)}, P_{X_j}^{\mathcal{C},do(X_i := \tilde{N}_i)}\right)$
6:         **end if**
7:     **end for**
8:     $\pi_{init} \leftarrow$ SORTRANKING($\mathbf{D}, \epsilon$)
9:     $\pi_{opt} \leftarrow$ LOCALSEARCH($\pi_{init}, \mathcal{S}, \epsilon, D, \mathcal{I}, P_X^{\mathcal{C},(\emptyset)}, \mathcal{P}_{int}$)
10:     **return** $\pi_{opt}$
11: **end procedure**

---

---

**Algorithm 2** Finding an initial solution

---

1: **procedure** SORTRANKING($\mathbf{D}$: matrix of distances, $\epsilon$)
2:     $G \leftarrow$ empty directed graph
3:     $sorted\_edges \leftarrow$ sort edges in the $d \times d$ matrix of distances $\mathbf{D}$ in descending order
4:     **for** each edge $(i,j)$ in $sorted\_edges$ **do**
5:         **if** there is no path from $j$ to $i$ in $G$ and $\mathbf{D}(i,j) > \epsilon$ **then**
6:             Add edge from $i$ to $j$ to $G$
7:         **end if**
8:     **end for**
9:     $\pi_{init} \leftarrow$ topological_order($G$)
10:     **return** $\pi_{init}$
11: **end procedure**

---

---

**Algorithm 3** Local search optimization

---

1: **procedure** LOCALSEARCH($\pi_{init}, \mathcal{S}$: score function of eq. (1), $\epsilon$, $D$: statistical distance function, $\mathcal{I}$: index set of intervened variables, $P_X^{\mathcal{C},(\emptyset)}$: observational distributions, $\mathcal{P}_{int}$: interventional distributions)
2:     $\pi_{current} \leftarrow \pi_{init}$
3:     $S_{current} \leftarrow \mathcal{S}(\pi_{current}, \epsilon, D, \mathcal{I}, P_X^{\mathcal{C},(\emptyset)}, \mathcal{P}_{int})$
4:     **while** True **do**
5:         $\Pi_{candidates} \leftarrow \Sigma(\{\pi_{current}\})$
6:         $\pi_{next} \leftarrow$ None
7:         **for** each $\pi_{candidate}$ in $\Pi_{candidates}$ **do**
8:             $S_{candidate} \leftarrow \mathcal{S}(\pi_{candidate}, \epsilon, D, \mathcal{I}, P_X^{\mathcal{C},(\emptyset)}, \mathcal{P}_{int})$
9:             **if** $S_{candidate} > S_{current}$ **then**
10:                 $\pi_{next} \leftarrow \pi_{candidate}$
11:                 $S_{current} \leftarrow S_{candidate}$
12:                 break
13:             **end if**
14:         **end for**
15:         **if** $\pi_{next}$ is None **then**
16:             break
17:         **end if**
18:         $\pi_{current} \leftarrow \pi_{next}$
19:     **end while**
20:     **return** $\pi_{current}$
21: **end procedure**

---

# D  WASSERSTEIN DISTANCE

The $\mathcal{W}_p$ $p$-Wasserstein distance between two distributions is defined as follows (Villani et al., 2009):

**Definition 2.** The $p$-Wasserstein distance $\mathcal{W}_p$ between two probability measures $P$ and $Q$ on $\mathbb{R}^d$ with finite $p$-moments, $p \in [1, \infty)$,

$$W_p(P, Q) = \inf_{\gamma \in \Gamma(P,Q)} \left( \int_{X \times X} \|x - y\|^p d\gamma(x, y) \right)^{\frac{1}{p}} \tag{4}$$

where $\Gamma(P, Q)$ is the set of joint probability measure on $\mathbb{R}^d \times \mathbb{R}^d$ with marginal $P$ and $Q$.

In this work, we use $p = 1$, and we have $d = 1$ as we only compute the distance between the marginal distribution of each variable. Let $P_n$ and $Q_n$ be distributions of samples from $P$ and $Q$ of size $n$. We have that the distance $W_p(P_n, Q_n)$ tends to $W_p(P, Q)$ as $n \to +\infty$. Furthermore, for $d \leq 2$, Ajtai et al. (1984) show that:

$$\mathbb{E}|W_p(P_n, Q_n) - W_p(P, Q)| = \mathcal{O}\left( \frac{\sqrt{\ln n}}{\sqrt{n}} \right). \tag{5}$$

# E  DETAILS OF EMPIRICAL EVALUATION

## E.1  LINEAR AND RFF DOMAIN

Each causal variable $x_j$ is defined with respect to its parents $x_{\mathrm{pa}(j)}$ through the equation:

$$x_j \leftarrow f_j(x_{\mathbf{Pa}_j^{\mathcal{G}}}, \epsilon_j) = f_j(x_{\mathbf{Pa}_j^{\mathcal{G}}}) + h_j(x_{\mathbf{Pa}_j^{\mathcal{G}}})\epsilon_j \tag{6}$$

Here, $\epsilon_j$ represents additive, potentially heteroscedastic noise, where the noise scale $h_j(x_{\mathrm{pa}(j)})$ is modeled as:

$$h_j(x) = \log(1 + \exp(g_j(x))) \tag{7}$$

with $g_j(x)$ being a nonlinear function. For the heteroscedastic case, we use random Fourier features with length scale 10.0 and output scale of 2.0.

To simulate interventions, the value of the intervened variable is set to fixed constant that is drawn from a signed Uniform distribution between 1.0 and 5.0.

### E.1.1  DOMAIN-SPECIFIC MODELLING

- **LINEAR Domain:** The causal functions $f_j$ are linear transformations:

$$f_j(x_{\mathbf{Pa}_j^{\mathcal{G}}}) = w_j^\top x_{\mathbf{Pa}_j^{\mathcal{G}}} + b_j \tag{8}$$

  where $w_j$ and $b_j$ are independently sampled for each transformation. As in , we sample $w_j$ from a signed Uniform distribution between 1 and 3, and $b_j$ from a Uniform distribution between $-3$ and 3.

- **RFF Domain:** Each $f_j$ is drawn from a Gaussian Process (GP) approximated using random Fourier features:

$$f_j(\mathbf{x}_{\mathbf{Pa}_j^{\mathcal{G}}}) = b_j + c_j\sqrt{\frac{2}{M}} \sum_{m=1}^{M} \alpha^{(m)} \cos\left( \frac{1}{\ell_j} \omega^{(m)} \cdot \mathbf{x}_{\mathbf{Pa}_j^{\mathcal{G}}} + \delta^{(m)} \right) \tag{9}$$

  where $\alpha^{(m)} \sim \mathcal{N}(0, 1)$, $\omega^{(m)} \sim \mathcal{N}(0, \mathbf{I})$, and $\delta^{(m)} \sim \mathrm{Uniform}(0, 2\pi)$, and the parameters $b_j$, $c_j$, and the length scale $\ell_j$ are sampled independently. The parameter $M$ is set to 100. The length scale $\ell_j$ is drawn from a Uniform distribution between 7.0 and 10.0, the output scale $c_j$ from a Uniform distribution between 10.0 and 20.0 and the bias $b_j$ from a Uniform distribution between $-3$ and 3.

## E.2  SIMULATION OF SINGLE-CELL GENE EXPRESSION DATA

As in Lorch et al. (2022), we use the SERGIO simulator by Dibaeinia & Sinha (2020) to generate realistic high-throughput scRNA-seq data, implemented in the AVICI package by Lorch et al. (2022).

### E.2.1 SIMULATION PROCESS WITH SERGIO

SERGIO generates gene expression data as snapshots from the steady state of a dynamical system governed by the chemical Langevin equation. The system's evolution is influenced by master regulators (MRs) with fixed rates and non-linear interactions among genes defined by the causal graph $G$. Cell types emerge from variations in MR rates, contributing to biological system noise and type-specific expression profiles.

### E.2.2 PARAMETERS FOR SIMULATION

To simulate $c$ cell types (5) across $d$ genes, SERGIO requires parameters such as interaction strengths $k$ (Uniform between 1.0 and 5.0), MR production rates $b$ (Uniform between 1.0 and 3.0), Hill coefficients $\gamma$ (2.0), decay rates $\lambda$ (0.8), and stochastic noise scales $\zeta$ (1.0). These are selected within recommended ranges to ensure realistic simulation. The values in parenthesis represents the parameter distributions we used. We did not simulate technical noise.

Intervention on a variable corresponds to knocking out a gene, that is, the expression of the gene is kept at 0.

### E.3 SIMULATION OF NEURAL NETWORK-BASED DATA FOR CAUSAL DISCOVERY

In the context of simulating data for causal discovery, we construct a domain where the data is generated by a set of random fully connected neural networks. These networks serve as the backbone for defining the observational conditional distributions within our simulated environment.

### E.3.1 NEURAL NETWORK SPECIFICATION

Each neural network is a multilayer perceptron (MLP) with a single hidden layer consisting of 10 neurons. Formally, the MLPs are functions $\text{MLP} : \mathbb{R}^j \to \mathbb{R}^1$, equipped with ReLU activation functions, mapping a $j$-dimensional input to a scalar output. The output of each MLP represents the mean $\mu$ of the conditional distribution for the corresponding variable:

$$p_j(x_j | x_{\mathbf{Pa}_j^{\mathcal{G}}}) \sim \mathcal{N}(\mu = \text{MLP}(x_{\mathbf{Pa}_j^{\mathcal{G}}}), \sigma = 1.0). \tag{10}$$

### E.3.2 INTERVENTIONAL DATA GENERATION

For the simulation of interventional data, the procedure modifies the distribution of intervened nodes. Instead of being determined by the MLP, the distribution of an intervened node is fixed to a marginal normal distribution:

$$p_j(x_j | \text{do}(x_j)) \sim \mathcal{N}(2, 1.0). \tag{11}$$

This alteration simulates the effect of an intervention in the data generation process.

## F ADDITIONAL EXPERIMENTS

### F.1 SIMULATIONS

Additional results for the simulation evaluations. Figure 3 shows the performance of Intersort for scale-free networks of size 5 and 30. Figures 4 and 5 show the performance of SORTRANKING on large scale settings of 1000 and 2000 variables, for both scale-free and Erdos-Renyi networks. Figure 6 presents a possible heuristic for selecting variables to intervene on, and comparing to a random selection.

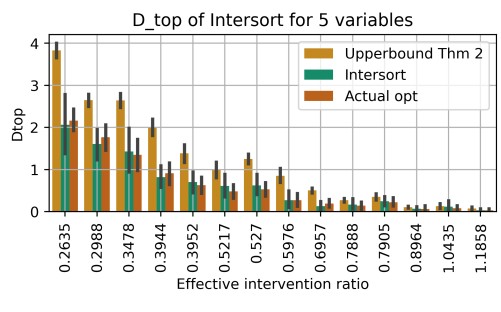

(a) Simulation with 5 variables

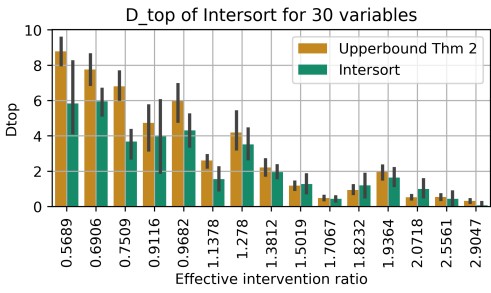

(b) Simulation with 30 variables

Figure 3: Simulation and comparison between the bounds of theorem 4 for scale-free networks, and between Intersort and the exact $\pi_{opt}$. For each setting, we draw 20 graphs per setting following a Barabasi-Albert scale-free distribution, with average edge per variable in $\{1, 2, 3\}$. A setting is the tuple $(p_{int}, p_e)$, where $p_e = \frac{2E(\#edges)}{d(d-1)}$. Then, for each graph, we run the algorithm on 10 configurations, where each configuration corresponds to a draw of the targeted variables following $p_{int}$. For both 5 and 30 variables, we have $p_{int} \in \{0.25, 0.33, 0.5, 0.66, 0.75\}$. The settings are ordered on the x-axis following what we call the effective intervention ratio $\frac{p_{int}}{\sqrt{p_e}}$. We observe that the error is approximately monotonic when ordered by the effective intervention ratio.

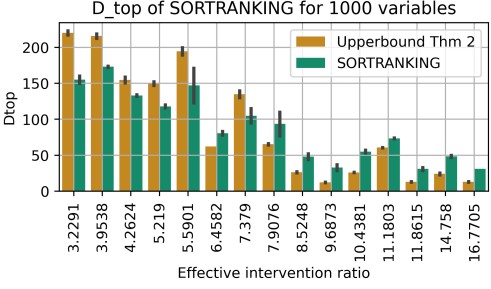

(a) Simulation with 1000 variables

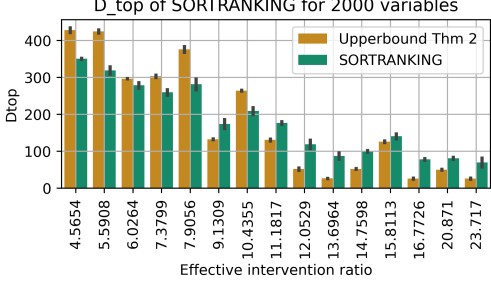

(b) Simulation with 2000 variables

Figure 4: Simulation and comparison between the bounds of theorem 4 for scale-free networks and SORTRANKING. For each setting, we draw 2 graphs per setting following a Barabasi-Albert scale-free distribution, with average edge per variable in $\{1, 2, 3\}$. A setting is the tuple $(p_{int}, p_e)$, where $p_e = \frac{2E(\#edges)}{d(d-1)}$. Then, for each graph, we run the algorithm on 1 configuration, where each configuration corresponds to a draw of the targeted variables following $p_{int}$. For both 5 and 30 variables, we have $p_{int} \in \{0.25, 0.33, 0.5, 0.66, 0.75\}$. The settings are ordered on the x-axis following what we call the effective intervention ratio $\frac{p_{int}}{\sqrt{p_e}}$. Even though the objective can be approximately solved at scale, we observe room for improvement as the performance of SORTRANKING is above the upper-bound in many settings.

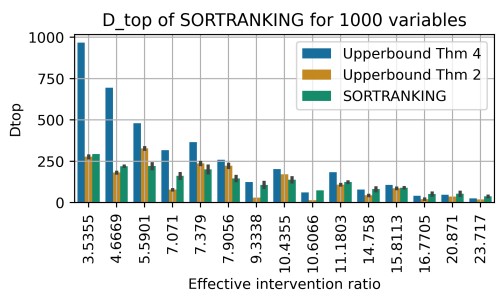 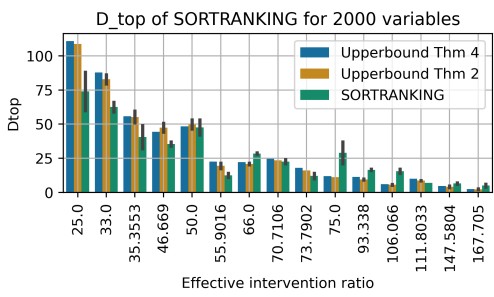

(a) Simulation with 1000 variables      (b) Simulation with 2000 variables

Figure 5: Simulation and comparison between the two bounds and SORTRANKING. For each setting, we draw 2 graphs per setting, where a setting is the tuple $(p_{int}, p_e)$. Then, for each graph, we run the algorithm on 1 configurations, where each configuration corresponds to a draw of the targeted variables following $p_{int}$. For both 5 and 30 variables, we have $p_{int} \in \{0.25, 0.33, 0.5, 0.66, 0.75\}$. In the 1000 variables setting on the left, we have $p_e \in \{0.005, 0.002, 0.001\}$. In the 20000 variables settings on the right, we have $p_e \in \{0.0001, 0.00005, 0.00002\}$. The settings are ordered on the x-axis following what we call the effective intervention ratio $\frac{p_{int}}{\sqrt{p_e}}$. Even though the objective can be approximately solved at scale, we observe room for improvement as the performance of SORTRANKING is above the upper-bound in many settings.

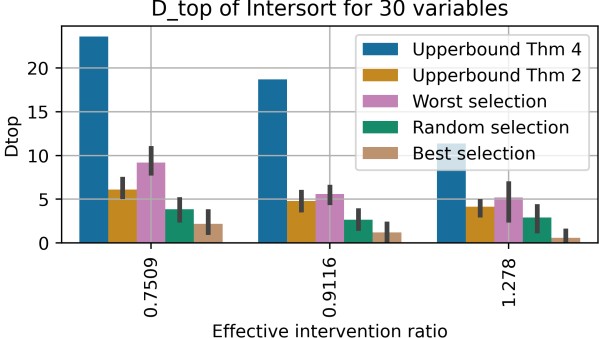

Figure 6: Simulation of scale free networks with average edge per variable in $\{1, 2, 3\}$ for 30 variables. We report the performance of Intersort with various intervention selection policies. We evaluate three policies: selecting the 10 variables with the most children, selecting 10 variables at random, and selecting the 10 variables with the fewest children. This corresponds to a setting with $30\%$ of intervened variables. We can thus hypothesize that a potential approach for intervention selection is to estimate the connectivity of the non-intervened variables.

## F.2 EMPIRICAL DATA

Results for the empirical evaluation for the smaller setting of 10 variables (fig. 7). Figure 8 shows the performance of SORTRANKING for 100 variables. Figure 9 analyzes the empirical difference in training time for the baselines on the neural network data type.

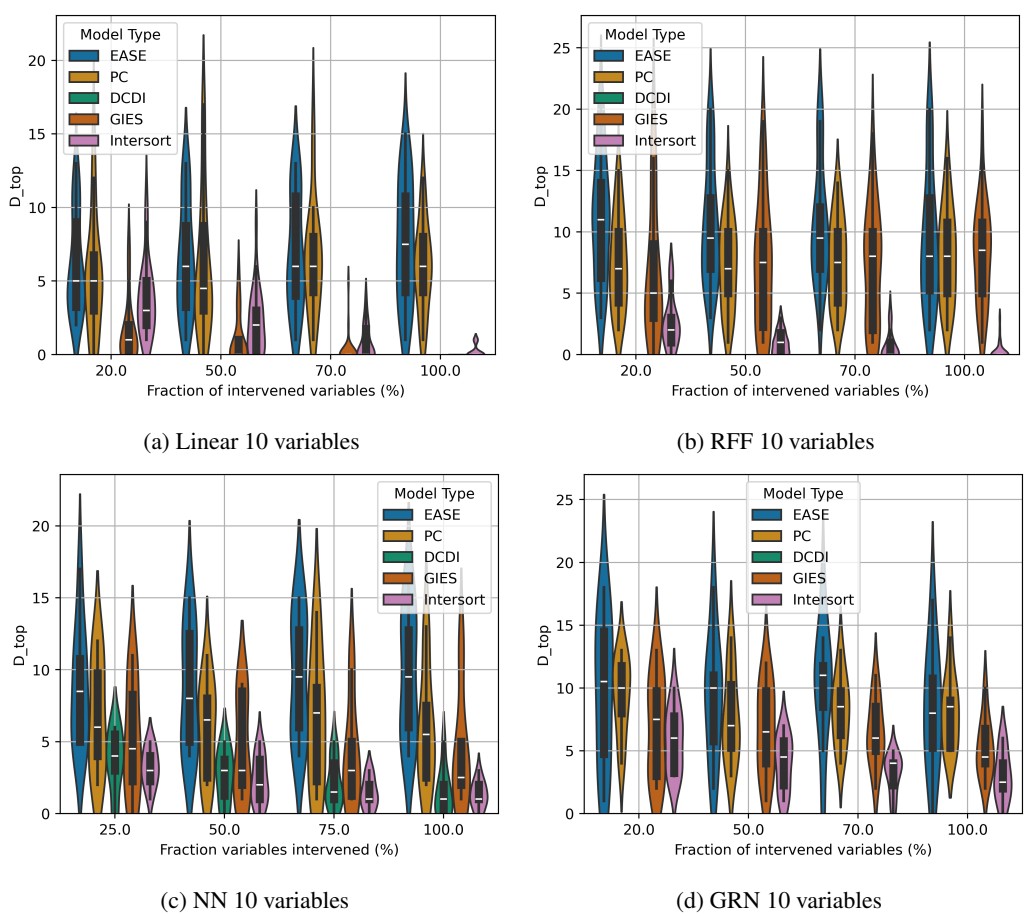

(a) Linear 10 variables

(b) RFF 10 variables

(c) NN 10 variables

(d) GRN 10 variables

Figure 7: Comparison of the performance of the baselines and of our model INTERSORT across diverse data domains as presented, for 10 variables. The x-axis corresponds to the fraction of variables that have been targeted by an intervention. The y-axis is the performance of causal ordering prediction as measured by the $D_{top}$ metric (see section 3).

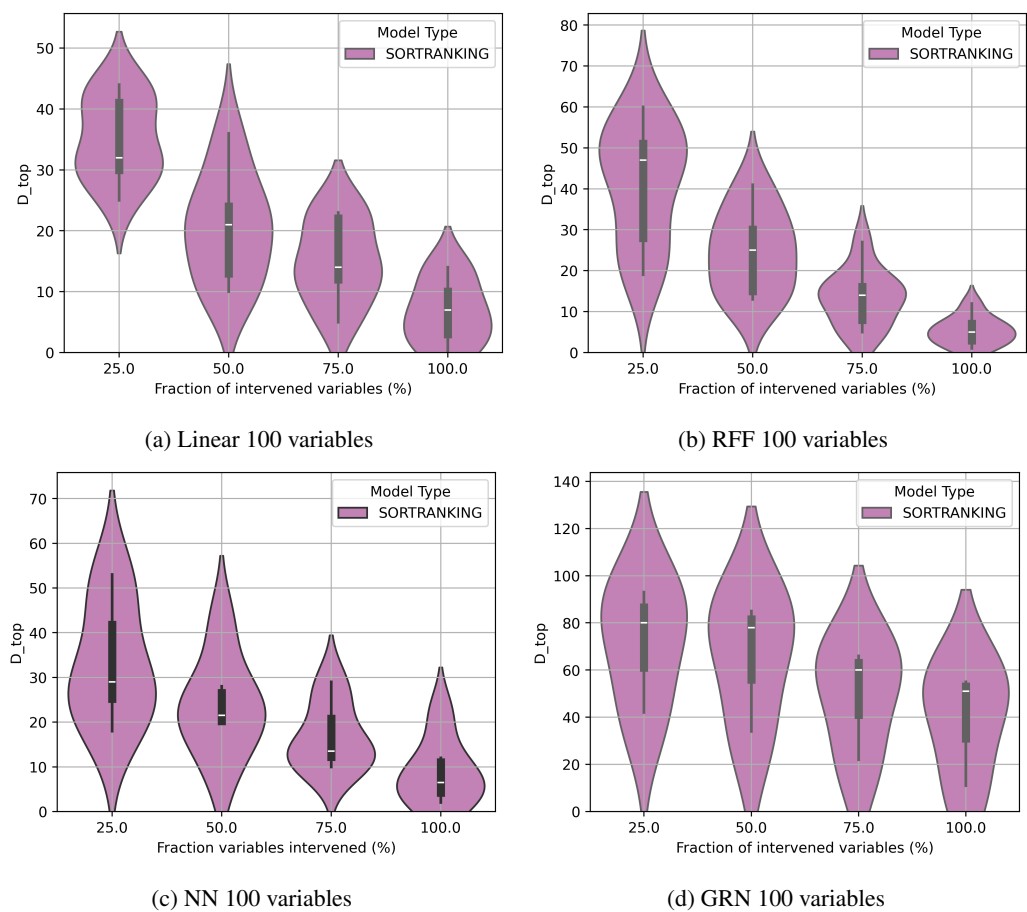

(a) Linear 100 variables

(b) RFF 100 variables

(c) NN 100 variables

(d) GRN 100 variables

Figure 8: Performance of our model SORTRANKING across diverse data domains as presented (linear, RFF, NN and GRN data), for 100 variables. The x-axis corresponds to the fraction of variables that have been targeted by an intervention. The y-axis is the performance of causal ordering prediction as measured by the $D_{top}$ metric (see section 3, lower is better).

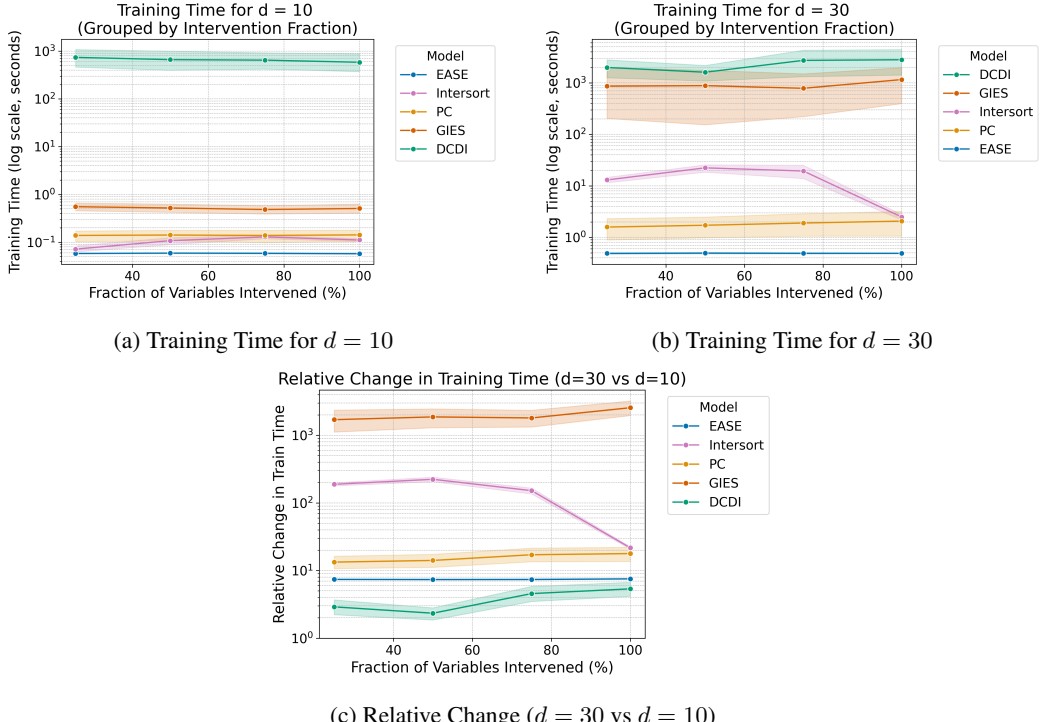

(a) Training Time for $d = 10$

(b) Training Time for $d = 30$

(c) Relative Change ($d = 30$ vs $d = 10$)

Figure 9: Training time analysis across different configurations for the NN data type. Plots (a) and (b) show the training time on a logarithmic scale for $d = 10$ and $d = 30$, respectively. Plot (c) shows the relative change in training time between $d = 10$ and $d = 30$, also on a logarithmic scale.

### F.3 QUALITATIVE EXAMPLE ON REAL-WORLD DATA

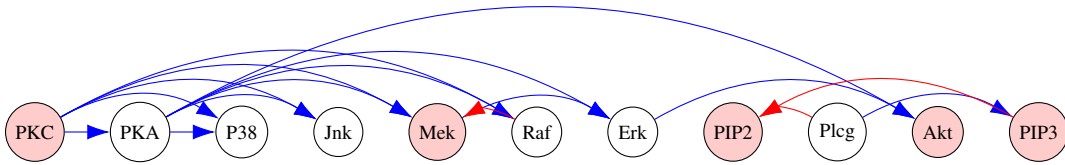

Figure 10: Output of Intersort on the flow cytometry data set of the Sachs et al. (2005) dataset, which measures the level of expression of phosphoproteins and phospholipids in human cells. Out of the 11 proteins, 5 (in red) where intervened on using reagents to activate or inhibit the measured proteins. The dataset comprises 5846 measurements, of which 1755 measurements are observational, and 4091 measurements are from the five different single node interventions. We here compare the predicted order of Intersort, using the Wasserstein distance and an epsilon value of 1.5, to the consensus true network consisting of 17 edges (Sachs et al., 2005). Intersort achieves a $D_{top}$ of 3, where we color the 3 edges in the wrong direction in red.

## G COMPUTATIONAL RESOURCES

The simulations of fig. 1 were run on an Apple M1 Macbook pro. The experiments for figs. 2 and 7 were run on a cluster with 20 CPUs, 16Gb of memory per CPU.

