# OpenReview forum: "Deriving Causal Order from Single-Variable Interventions: Guarantees & Algorithm"
_ICLR.cc/2025/Conference — ICLR 2025 Poster_

### Official Review · Reviewer_mrVM · 2024-10-16

**Soundness:** 3
**Presentation:** 3
**Contribution:** 3
**Rating:** 8
**Confidence:** 3

**Summary:**

The paper proposes a new method to infer the causal order of an (unknown) causal graph from interventional data. For this purpose, the authors introduce a new assumption (interventional faithfulness), a theoretically grounded score on causal orders, and a practical algorithm called intersort approximating the proposed score. Intersort is evaluated on simulated data.

**Strengths:**

- The proposed method is the first to estimate the causal order from interventional data
- The paper contains several novel and theoretically grounded contributions, including the causal score as well as the algorithm to approximate it
- Promising empirical results on simulated data

**Weaknesses:**

- As common in causal discovery, the method relies on generally untestable assumptions (e.g., epsilon interventional faithfulness)
- The method relies on estimating distributional distances (e.g., Wasserstein distance) which can be statistically challenging
- No experiments using real-world data are provided. I understand that benchmarking methods are challenging due to unknown causal ground truth, however, I think it would be nice to sketch an application of the method on real-world data and potentially obtain insights
- The related work on causal discovery could be expanded

Minor
- Appendix B seems to be missing
- There is some weird formatting on page 16 (Appendix C and D intersect)
- There is no text in Appendix F.1/ F.2 so it is hard to understand which figures belong to which section. Also, Appendix G intersects with F2

**Questions:**

- What would be an application in which one would only be interested in inferring the causal order and not the full graph/Markov equivalence class?
- How exactly is the algorithm implemented? E.g., which p-Wasserstein distance is chosen? Which algorithm is used to compute the distance?

---

> ### Author Response · Authors · 2024-11-23
> **Authors response to Official Review of Submission6423 by Reviewer mrVM**
>
> >“No experiments using real-world data are provided. I understand that benchmarking methods are challenging due to unknown causal ground truth, however, I think it would be nice to sketch an application of the method on real-world data and potentially obtain insights”
>
> We also appreciate the importance of applicability on real-world data and had this aspect in mind in the development of the method. As a proof of concept, we additionally ran Intersort on the Sachs 2005 dataset, which is a biological dataset consisting of 11 variables and 5 interventions. We compare the predicted order to the consensus true graph in Appendix F3, Figure 10 of the revised manuscript.
>
> >“The related work on causal discovery could be expanded”
>
> Thank you for this suggestion. We are happy to extend the related work section in the camera-ready version to provide a more comprehensive overview of the current landscape in causal discovery. This will ensure that our contributions are clearly contextualized within the broader body of work.
>
> >“Appendix B seems to be missing”
>
> >“There is some weird formatting on page 16 (Appendix C and D intersect)”
>
> >“There is no text in Appendix F.1/ F.2 so it is hard to understand which figures belong to which section. Also, Appendix G intersects with F2”
>
> Thank you for pointing out the formatting issues in the appendices. We have fixed the formatting in the revised version of our manuscript.
>
> >“What would be an application in which one would only be interested in inferring the causal order and not the full graph/Markov equivalence class?”
>
> While it's true that comprehensive causal graphs can provide answers to questions regarding influences and interactions, we argue—and demonstrate to some extent in our paper—that focusing on learning the causal order offers a more robust and realistic approach under many circumstances. Causal order allows for understanding which elements in a system are influential, thereby offering a practical framework for targeting interventions and exploring causal mechanisms.
>
> In fields like genetics or systems biology, knowing the causal order can help differentiate between potential symptoms and root causes of a disease. This can streamline research by highlighting the most impactful variables to investigate further.
>
> Additionally, the process of uncovering causal order tends to be less sensitive to the complexities and data requirements inherent in learning an entire causal graph, making it a more attainable goal given partial intervention data.
>
> Thus, while causal graphs have their place, focusing on causal order provides a practical, robust alternative that supports meaningful research and application, as partially evidenced by the findings and approaches discussed in our work.
>
> >“How exactly is the algorithm implemented? E.g., which p-Wasserstein distance is chosen? Which algorithm is used to compute the distance?”
>
> We utilize the [Wasserstein-1 distance from the scipy library](https://docs.scipy.org/doc/scipy/reference/generated/scipy.stats.wasserstein_distance.html) to compute the statistical distance between distributions, with the default parameters.
>
> Reference:
>
> Karen Sachs, Omar Perez, Dana Pe’er, Douglas A Lauffenburger, and Garry P Nolan. Causal protein-signaling networks derived from multiparameter single-cell data. Science, 308(5721): 523–529, 2005

---

> > ### Comment · Reviewer_mrVM · 2024-11-25
> > **Rebuttal acknowledged**
> >
> > I appreciate the clarifications and additional experiments on real-world data. I have increased my score accordingly.

---

### Official Review · Reviewer_nPs4 · 2024-10-28

**Soundness:** 2
**Presentation:** 3
**Contribution:** 2
**Rating:** 6
**Confidence:** 3

**Summary:**

This paper proposed INTERSORT, an algorithm for causal discovery with interventional data, to find a casual relationship with distribution change after intervention. Theoretically, the upper bound of the error is provided for the optimal causal order. Experimental results on simulated and real-world datasets verify the effectiveness of the proposed method.

**Strengths:**

The score of causal order is designed to identify causal relationships with the help of distribution change engendered by intervention. Theoretically, the upper bound of the expected error is provided for the optimal causal order.

**Weaknesses:**

1. If setting the threshold following the Remark in Line 203, then it means every pair with distribution change after perturbation will be considered. Why not directly set it to 0, which means detecting distribution change directly?
2. The selection processes also have a large distance, however, this is not a causal relationship. Moreover, selection is common in gene regulatory networks. In your method, how to guarantee the ones you find are causal relations?
3. In eq. (1), what does the second part mean? I scratch my head to understand it. Could you provide descriptions of the symbols used in the equation?
4. In Algorithm 1, how many kinds of permutations will be considered. Usually, it will be non-traversable with an increase in the number of variables.
5. In the part of Distribution with intervention, ' the structural assignment is replaced by a new random variable independent of the parents' means the hard intervention right? Moreover, the author also mentioned that distribution after intervention can be accessed, do you have assumptions about the distribution of intervention? and why you need the distribution needs to be accessed.

**Questions:**

1. If setting the threshold following the Remark in Line 203, then it means every pair with distribution change after perturbation will be considered. Why not directly set it to 0, which means detecting distribution change directly?
2. The selection processes also have a large distance, however, this is not a causal relationship. Moreover, selection is common in gene regulatory networks. In your method, how to guarantee the ones you find are causal relations?
3. In eq. (1), what does the second part mean? I scratch my head to understand it.
4. In Algorithm 1, how many kinds of permutations will be considered. Usually, it will be non-traversable with an increase in the number of variables.
5. In the part of Distribution with intervention, ' the structural assignment is replaced by a new random variable independent of the parents' means the hard intervention right? Moreover, the author also mentioned that distribution after intervention can be accessed, do you have assumptions about the distribution of intervention? and why you need the distribution needs to be accessed.

---

> ### Author Response · Authors · 2024-11-23
> **Authors Response to Official Review of Submission6423 by Reviewer nPs4**
>
> We appreciate your thorough and thoughtful review highlighting both strengths and areas for improvement in our manuscript. Your feedback has been invaluable in pinpointing where additional clarity and detail are needed. We have addressed your questions and concerns, especially regarding threshold choices, the assurance of causal relationships, and the clarity of our scoring methodology. We are grateful for your insights, which have guided us in enhancing the robustness and clarity of our work.
>
> >“If setting the threshold following the Remark in Line 203, then it means every pair with distribution change after perturbation will be considered. Why not directly set it to 0, which means detecting distribution change directly?”
>
> In practical scenarios where we work with non-asymptotic data, the calculated statistical distances are almost always non-zero, even in the absence of a causal link. This is due to the presence of noise and sampling variability inherent in real-world data.
> Our ϵ-interventional faithfulness assumption is useful precisely because it incorporates a significance threshold, $(\epsilon)$, to distinguish between true causal effects and these non-zero distances that arise from statistical noise or minor fluctuations. Setting the threshold to zero would not effectively differentiate meaningful changes from these artifacts, leading to potential false positives.
> In practice, it's essential to set $(\epsilon)$ higher than the observed distance when no causal link exists. This careful calibration helps ensure that the detected changes are significant and representative of genuine causal relationships, rather than being driven by random variations. We appreciate your question as it allows us to clarify the practical importance of setting an appropriate threshold in our methodology.
>
>
> >“The selection processes also have a large distance, however, this is not a causal relationship. Moreover, selection is common in gene regulatory networks. In your method, how to guarantee the ones you find are causal relations?”
>
> Thank you for raising the concern about distinguishing causal relationships from selection effects, especially within the context of gene regulatory networks. In our work, we focus on using interventional data, which allows us to more effectively disambiguate confounded dependencies that might be present in observational data alone. By actively intervening on specific variables and observing changes, our approach can differentiate true causal links from mere correlations that might have been influenced by hidden confounders or selection effects.
> Moreover, we have added a section to the manuscript that discusses the role of confounders and how our method can be extended to derive causal order within subsets of variables. This section leverages the framework of Acyclic Directed Mixed Graphs (ADMGs) to account for potential confounders, providing a more robust understanding of causal relationships even when some variables are unobserved or affected by hidden factors. This enhancement helps ensure that the causal relationships identified through our method are genuine and not artifacts of confounded patterns in the data.
> Your question highlights the importance of using interventional approaches to uncover true causal structures, and we appreciate the opportunity to clarify how our methodology addresses these critical aspects.
>
> >“In eq. (1), what does the second part mean? I scratch my head to understand it. Could you provide descriptions of the symbols used in the equation?”
>
> The coefficient $(O(d))$ in the second term of the score is a technical modification intended to emphasize the influence of causal effects that are greater than the significance threshold $(\epsilon)$. To provide some intuition: the terms in the summation where the statistical distance is smaller than $(\epsilon)$ contribute negatively to the score by an order of magnitude roughly proportional to $(d)$ for each intervention. This inflation is necessary to ensure that the score strongly favors orderings where $(\pi(i) < \pi(j))$ for any pair $((i, j))$ where the distance exceeds $(\epsilon)$. By weighting the significant effects more heavily, the scoring function guides the optimization towards a causal order that better reflects meaningful causal relationships in the data.
> In revising the presentation of this scoring method in our paper, we've included a more detailed explanation to provide better insight into this construction.

---

> > ### Author Response · Authors · 2024-11-23
> >
> > >“In Algorithm 1, how many kinds of permutations will be considered. Usually, it will be non-traversable with an increase in the number of variables.”
> >
> > You are indeed correct that the number of permutations can become non-traversable as the number of variables increases. In practice, our approach manages this complexity by utilizing the SORTRANKING component to provide an initial ordering based on available single-variable interventions and statistical distances. This initial step significantly reduces the permutation space that needs to be explored during the LOCALSEARCH phase.
> > To further enhance scalability, the LOCALSEARCH phase does not exhaustively explore all permutations but instead employs optimization strategies focused on local improvements. This allows us to efficiently search for a near-optimal causal order without the necessity of evaluating all possible permutations.
> > We recognize that addressing scalability is a critical challenge, and it is an important direction for future developments. Incorporating heuristic or probabilistic techniques to guide the search process could further enhance the algorithm's efficiency in larger settings. To stress the importance of this aspect, we have revised the conclusion to mention scalability as a direction of future work.
> >
> > >“In the part of Distribution with intervention, ' the structural assignment is replaced by a new random variable independent of the parents' means the hard intervention right? Moreover, the author also mentioned that distribution after intervention can be accessed, do you have assumptions about the distribution of intervention? and why you need the distribution needs to be accessed.”
> >
> > You are correct in noting that when we mention that "the structural assignment is replaced by a new random variable independent of the parents," it refers to a type of intervention that can certainly include hard interventions. However, our method is not limited to hard interventions; we also consider stochastic interventions, where the intervening variable may follow a random distribution.
> >
> > The key assumption we make about interventions is that they are perfect, meaning they fully determine the value of the variable being intervened upon without being influenced by parental nodes in the causal graph.
> >
> > The assumption about interventional distributions is encapsulated within our ϵ-interventional faithfulness criteria. This ensures that the interventions result in meaningful changes that are detectable in the marginal distributions of the variables. Access to the intervened distribution is necessary for computing statistical distances, which form the basis of determining causal relationships in our approach.

---

> > > ### Comment · Reviewer_nPs4 · 2024-11-27
> > >
> > > I appreciate the authors' response to my questions. I am clear on most concerns now. For Q (2), it is clear that the latent confounder will not have a distribution change after intervention. However, the selection structures will. If your algorithm can not handle this, the statements about the structure conditions should be mentioned to clarify this.

---

> > > > ### Author Response · Authors · 2024-11-28
> > > >
> > > > We thank the reviewer for acknowledging our efforts in addressing their concerns and for their further comment on the effect of the selection process in our algorithm. This insightful remark helps to better illuminate the potential limitations of our approach. While we are not entirely certain of the exact meaning of "selection structure" in this context, we interpret it in a few possible ways and provide clarifications accordingly:
> > > >
> > > > 1. **Selection of Variables in the Observational Data**: If "selection structure" refers to the selection of a subset of variables from the full causal system (e.g., gene subsets in genomics), we note that this scenario is now explicitly addressed in the revised manuscript. In the newly added section, we leverage the framework of Acyclic Directed Mixed Graphs (ADMGs) to account for the challenges posed by latent variables and selection processes. This addition ensures a more comprehensive treatment of cases involving partial variable selection.
> > > > 2. **Selection Bias in Data Collection**: If "selection structure" pertains to biases in the sampling process (e.g., preferential inclusion of certain data points or experiments), we acknowledge that our algorithm does not explicitly handle such biases. However, these effects can often be mitigated by ensuring diverse and unbiased intervention datasets, which is an important consideration in practical applications. We have clarified this point in the revised manuscript (end of the confounder section 5).
> > > > 3. **Changes in Structural Relationships Due to Selection Post-Intervention**: If "selection structure" refers to changes in latent-variable-induced dependencies (e.g., colliders or confounders) after interventions, we assume that latent confounders remain stable across conditions. We have further emphasized the implications of this assumption in the revised text to ensure clarity (end of the confounder section 5).
> > > >
> > > > We hope these clarifications address your concern. If there is another aspect of "selection structure" that we have not covered or if our clarifications are insufficient, we would greatly appreciate further elaboration so that we can ensure all your concerns are fully addressed.

---

### Official Review · Reviewer_PzaP · 2024-10-31

**Soundness:** 4
**Presentation:** 3
**Contribution:** 3
**Rating:** 8
**Confidence:** 3

**Summary:**

Learning the topological ordering of the causal graph behind a set of variables is useful in a variety of scientific domains. For instance, when gene expressions are correlated, the causal order can help discern the structure of the regulatory network. This paper proposes a novel approach for ordering variables using many single-variable interventions. Strong theoretical guarantees and diverse empirical evaluations are presented.

**Strengths:**

This paper presents an original solution to a significant problem: a method for directly learning the causal order using interventional data.
 * Although technically dense, the paper is surprisingly easy to read and well-organized.
 * The theoretical guarantees for optimality under a reasonable faithfulness assumption appear solid.
 * The approximation algorithm is computationally tractable and validated with a variety of benchmarks.

**Weaknesses:**

* The majority of the analysis assumes access to oracle statistical distances between interventional distributions. Little attention is paid to the estimation of these distances using samples, and how they affect the sorting algorithm.
 * The main score objective (Equation 1) is difficult to understand and not explained much.

Minor comments
 * Please define all the terms on like 152, like the noise variable $N_j$.

**Questions:**

Are there useful heuristics for choosing which variables to intervene on ($\mathcal{I}$)?

---

> ### Author Response · Authors · 2024-11-23
> **Authors Response to Official Review of Submission6423 by Reviewer PzaP**
>
> Thank you for your thorough and thoughtful review of our manuscript. We are pleased that you found our approach novel and our presentation clear.
>
> >“The majority of the analysis assumes access to oracle statistical distances between interventional distributions. Little attention is paid to the estimation of these distances using samples, and how they affect the sorting algorithm.”
>
> While the estimation of these distances using samples is an important consideration, it is outside the current scope of our paper. However, we recognize that developing a theoretical framework to better understand how factors such as the choice of distance metric, number of samples, and number of variables influence the sorting algorithm is indeed a fascinating direction for future research.
> In our work, we primarily use the Wasserstein distance owing to its empirical effectiveness in our experiments. Nevertheless, a broader analysis could explore different distance metrics and their robustness to sampling variability, potentially offering insights into optimizing distance metric choice based on specific data characteristics and constraints.
> We appreciate your insight, which highlights a valuable area for future investigation.
>
>
> >“The main score objective (Equation 1) is difficult to understand and not explained much.”
>
> We recognize the importance of clarity in conveying our scoring methodology and have taken steps to address this in the updated manuscript. We have revised the section detailing Equation 1 to provide a more intuitive explanation, including the rationale and significance of each component in the scoring function.
> These improvements aim to make the underlying logic of the score more accessible, ensuring that its role and impact within the framework are clearly understood.
>
> >“Please define all the terms on like 152, like the noise variable Nj.”
>
> Thank you for pointing out the need for clearer definitions of terms, such as the noise variable $(N_j)$. In response, we have revised the presentation of the Structural Causal Model (SCM) in our manuscript to ensure greater mathematical precision. This includes providing detailed definitions and explanations for all relevant terms, helping to clarify their roles and relationships within the models.
>
> >“Are there useful heuristics for choosing which variables to intervene on (I)?”
>
> In practical applications, strategically selecting interventions rather than choosing randomly can significantly enhance the method's data efficiency.
> One useful heuristic we propose is to target variables that are highly connected, focusing on those with a large number of descendants or ancestors. Such variables are likely to play pivotal roles in the causal structure, and intervening on them may provide substantial information about the causal order.
> To explore this concept, we conducted a small experiment, which we have included in Appendix F1, Figure 6. This experiment investigates the impact of different intervention policies in a scale-free graph with 30 nodes. We compared three strategies: selecting the 10 variables with the fewest children, choosing 10 variables at random, and selecting the 10 variables with the most children. Although this analysis is not entirely realistic as it leverages knowledge of the true graph, it demonstrates the potential benefits of informed intervention selection.
> We appreciate your question as it highlights an important practical consideration that can improve the effectiveness of causal discovery processes.

---

> > ### Comment · Reviewer_PzaP · 2024-11-26
> > **Acknowledgment**
> >
> > Thank you for your thoughtful response. In light of the other reviews and responses, I have chosen to maintain my positive assessment.

---

### Official Review · Reviewer_WsAd · 2024-11-01

**Soundness:** 3
**Presentation:** 4
**Contribution:** 4
**Rating:** 8
**Confidence:** 4

**Summary:**

The paper introduces the concept of "interventional faithfulness," which relies on comparisons between the marginal distributions of each causal variable across observational and interventional settings, and also develops a scoring function for ranking causal orders of variables. Based on this concept, the authors 1) provide strong theoretical guarantees on the optimum of the proposed score and 2) propose INTERSORT to infer causal order from datasets containing large amounts of single-variable interventions. INTERSORT outperforms existing methods in various simulations, demonstrating the potential of the new theory for advancing causal inference in domains like biology.

**Strengths:**

* The paper introduces a new definition of faithfulness with both theoretical guarantees and empirical results.
* It is well-written and clear.
* Numerical experiments are designed in a sensible manner that adequately supports the claims.
* This is an important and highly relevant contribution to the community, with developed theory that has the potential to further advance causality.

**Weaknesses:**

* It’s unclear how the findings would generalize to different settings, such as different distributions over random interventional variables, or in the case of having discrete causal variables.
* Empirical experiments are conducted on a limited set of underlying models.

**Questions:**

1. Are there scenarios where the proposed method cannot be applied? In other words, are there real-world systems that do not satisfy e-interventional faithfulness or its relaxations?
2. How could INTERSORT be extended to handle a large number of nodes? I presume for calculating distance, one could replace Wasserstein with e.g. MMD, but what would be the main bottleneck of the algorithm aside from the distance metric?
3. Could the authors elaborate on how the proposed framework could be extended to derive causal order over a subset of variables?
4. Do the authors have any intuition on how this framework might be used to develop better causal discovery algorithms?

---

> ### Author Response · Authors · 2024-11-23
> **Authors response to Official Review of Submission6423 by Reviewer WsAd**
>
> Thank you for your valuable feedback and for recognizing the contributions of our work. Your insights have been instrumental in refining our manuscript, and we've addressed your comments to clarify and enhance our research. We appreciate the opportunity to improve our paper further.
>
> >“It’s unclear how the findings would generalize to different settings, such as different distributions over random interventional variables, or in the case of having discrete causal variables.”
>
> We appreciate your interest in exploring the adaptability and robustness of our approach.
>
> Our methodology is designed to be broadly applicable across varied distributional settings due to the flexibility of the ϵ-interventional faithfulness assumption, which requires that interventions lead to significant changes in marginal distributions of variables. This assumption is intentionally less restrictive, allowing it to cover a wide range of structural causal models (SCMs). Importantly, our theoretical results accommodate any distribution for the interventional random variables, as well as discrete data, provided these meet the ϵ-interventional faithfulness criteria.
>
> >“Empirical experiments are conducted on a limited set of underlying models.“
>
> We acknowledge that the current empirical studies have limitations in terms of model diversity. We focused on certain common types like linear, Random Fourier Features, neural network, and single-cell data, which are representative but not exhaustive. Future work will investigate extensions to more diverse data types, including discrete variables, to empirically substantiate the theoretical flexibility.
> In cases where the data structure deviates significantly from our assumptions, particularly with discrete variables or unconventional interventional distributions impacting faithfulness, adjustments might be necessary. For instance, future extensions might include tailored statistical measures to cater to specific distributional characteristics or discrete data challenges.
>
> >“Are there scenarios where the proposed method cannot be applied? In other words, are there real-world systems that do not satisfy e-interventional faithfulness or its relaxations?”
>
> While the ϵ-interventional faithfulness assumption offers flexibility and covers a broad range of scenarios, there are indeed settings where our method might face challenges.
> One primary limitation is in systems with cyclic dependencies. Our framework is primarily designed for acyclic causal structures, and the presence of cycles could violate the assumptions necessary for accurately inferring causal order. This may result in a number of omitted interactions. Additionally, scenarios where interventions do not result in substantial changes to the marginal distributions, leading to weak or negligible causal effects, might not meet the ϵ-interventional faithfulness criteria effectively.
>
> These cases emphasize the importance of assessing structural characteristics and the strength of interventions when applying our method. We appreciate your question, as it highlights crucial considerations for ensuring the effective application of our approach in real-world systems.
>
> >“How could INTERSORT be extended to handle a large number of nodes? I presume for calculating distance, one could replace Wasserstein with e.g. MMD, but what would be the main bottleneck of the algorithm aside from the distance metric?”
>
> Scalability is indeed a crucial area for future work. While replacing the Wasserstein distance with a more computationally efficient metric like MMD can help reduce runtime, the primary bottleneck of INTERSORT lies in the LOCALSEARCH component.
> One immediate approach to address this is to rely solely on the SORTRANKING part of the algorithm, which, as evidenced in Figures 4, 5, and 8, can still provide meaningful results. To further enhance scalability, improvements in optimization techniques are necessary. Exploring advanced optimization strategies or heuristics that can streamline LOCALSEARCH without compromising accuracy is an avenue we plan to investigate in future work.
> We appreciate your question, which underscores the importance of enhancing the algorithm's efficiency for broader applicability.

---

> > ### Author Response · Authors · 2024-11-23
> >
> > >"Could the authors elaborate on how the proposed framework could be extended to derive causal order over a subset of variables?”
> >
> > This query prompted us to look deeper into this issue, leading to an extension of the paper with a section focusing on confounders.
> > In the revised manuscript, we introduce a section discussing how our framework accommodates scenarios involving latent confounders. This extension includes the use of Acyclic Directed Mixed Graphs (ADMGs) to model dependencies even when not all variables are directly observed. This setting also accomodates the case where we aim to apply our method only on a subset of variables. This result means that the method can be applied as is to a subset of variables, by only taking into account the interventions on variables that are in the subset.
> > Thank you for your insightful query, which helped drive this important enhancement to the paper.
> >
> > >“Do the authors have any intuition on how this framework might be used to develop better causal discovery algorithms?”
> >
> > Thank you for offering us the opportunity to elaborate on this aspect. We do see potential for our framework to contribute significantly to the development of better causal discovery algorithms. Our intuition is that leveraging ϵ-interventional faithfulness offers substantial advantages, and methods that integrate this inductive bias are likely to achieve improved performance, for example by orienting uncertain edges or by guiding the optimization process.
> > One straightforward way to utilize this framework is to employ the output of INTERSORT to either orient edges directly or serve as an initial solution in causal discovery processes. Additionally, our scoring methodology could be integrated into score-based methods to refine their effectiveness. While the path to applying this framework to differentiable discovery models is more complex, we believe it is feasible—potentially by adapting the score to be differentiable.

---

> > > ### Comment · Reviewer_WsAd · 2024-11-25
> > >
> > > I appreciate the authors for responding to my questions. After reviewing the other comments and considering the authors' responses, I will keep my positive score and recommend the acceptance of the paper.

---

### Official Review · Reviewer_R6EY · 2024-11-01

**Soundness:** 2
**Presentation:** 3
**Contribution:** 2
**Rating:** 5
**Confidence:** 4

**Summary:**

The authors considered the problem of recovering a causal order in structural causal models (SCM) under the causal sufficiency assumption (it seems that is the case but it is not clearly mentioned in the paper). They assumed that they have access to observational data and some single-variable interventional ones and used the changes observed in the marginal distributions of variables to recover a causal order. The authors introduced a notion of faithfulness (interventional faithfulness) and showed that a causal order can be recovered by maximizing a score (defined in (1)) if we have single-variable interventions on all the variables in the system. They proposed a heuristic algorithm (called INTERSORT) aiming to improve the score. The experimental results showed that in some specific settings, the proposed algorithm has a better performance compared to some of the previous work.

**Strengths:**

Originality/Quality: The authors asserted at the end of the Related Work section that they are proposing the first algorithm to infer the causal order from the interventional data. I do not think this is true. First, (Tian and Pearl, 2013) is one of the earliest works using the changes in marginal distributions due to intervention in inferring some orders among the variables in the system (See Section 4 there). Second, there are extensive works on recovering an equivalence class of models from the observational and interventional data (whether the intervention locations are known or not). A few works are cited in the paper (such as GIES and DCDI). These works can also provide some information about the causal orders that are encoded in the recovered equivalence class.

Clarity: The paper is generally well-written.

Significant: Based on what was mentioned, I think that the authors should carefully compare their methods with previous work (such as (Tian and Pearl, 2013) when $\epsilon=0$). Moreover, from the experimental results, it is not clear that the proposed algorithm indeed improves SOTA.

**Weaknesses:**

Comparison with previous work: I think there is no clear comparison with previous work (especially with (Tian and Pearl, 2013)) and discussion about the advantages of the current approach.

Theoretical result in a very limited setting: I think the assumption of having single-variable intervention on all the variables is very restrictive (what can we say in theory about the recovered causal order if a portion of variables are intervened on?). Moreover, the proposed algorithm is designed under the causal sufficiency assumption (which as far as I checked, is not clearly mentioned in the paper).

The notion of interventional faithfulness: I think this assumption is an extension of "influentially" in (Tian and Pearl, 2013). The connection to that definition is not discussed in the paper.

Theoretical guarantee about INTERSORT: Although the authors used the term "approximation algorithm" in line 359, there is no theoretical guarantee about the quality of the output of INTERSORT.

**Questions:**

1. What are the main differences between the current work and (Tian and Pearl, 2013) in terms of faithfulness assumption and methodology?

2. In lines 65-68, the authors are giving the advantage of knowing the order. Did they mean that the number of candidates is divided by 4 if we know that the target gene is in the middle of the causal order?

3. In eq. (1), in the second term, why is there a coefficient of $d$?

4. In Lemma 4, based on the chosen $p_e$, it seems that the graph is disconnected with high probability. Can the authors elaborate more on this?

5. In line 283, the authors argued that the expected error is growing with the order $O(d)$ and they mentioned that it is a strong guarantee. I think the upper bound on the expected error is $d^2$. Therefore, I am not sure this is indeed a strong result.

6. The pseudocode in Algorithm 1 is somehow useless. The most important parts (SORTRANKING and LOCACLSEARCH) are not described there.

7. Evaluating empirically does not imply the quality of the output of the algorithm in theory. I suggest removing the term "approximation algorithm" in line 359.

8. What is the computational complexity of the proposed algorithm? It is also good to compare the algorithms in terms of runtime empirically.

9. In Fig. 2, in some settings (such as NN 30 variables or GRN with a fraction of 23.33%), other methods have better performance. I think it is good to elaborate more on these cases in the experiment section.

---

> ### Author Response · Authors · 2024-11-23
> **Authors response to Official Review of Submission6423 by Reviewer R6EY**
>
> We thank you for your mindful review. We repond to your concerns point by point, especially around the comparison to previous work such as Tian and Pearl.
>
> >The authors asserted at the end of the Related Work section that they are proposing the first algorithm to infer the causal order from the interventional data. I do not think this is true. First, (Tian and Pearl, 2013) is one of the earliest works using the changes in marginal distributions due to intervention in inferring some orders among the variables in the system (See Section 4 there).
>
> >The notion of interventional faithfulness: I think this assumption is an extension of "influentially" in (Tian and Pearl, 2013). The connection to that definition is not discussed in the paper.”
>
> >What are the main differences between the current work and (Tian and Pearl, 2013) in terms of faithfulness assumption and methodology?“
>
> We appreciate the opportunity to further clarify and position our work in the context of existing literature. Below, we address your points regarding comparisons with Tian and Pearl:
> We acknowledge Tian and Pearl major contribution to inferring causal orders using interventional data.
> Our notion of interventional faithfulness extends Tian and Pearl’s 'influentiality' by  utilizing statistical distances and thresholds to establish causal relations more explicitly. This provides a clear pathway to introduce our score and simplifies its theoretical development.
> Regarding methodology, our score-based approach contrasts with Tian and Pearl's rule-based algorithm by enabling the application of various optimization techniques, enhancing scalability and future-proofing the methodology for broader application in larger networks.
> Additionaly, we provide extended theoretical results, including error bounds applicable even when intervention coverage is partial.
>
> In light of these discussions, we have revised the related work section in our paper to more accurately reflect our contributions and context within the existing causal inference literature.
> We hope this response satisfactorily addresses your valuable feedback and contextualized our work better in the existing literature.
>
> >Theoretical result in a very limited setting: I think the assumption of having single-variable intervention on all the variables is very restrictive (what can we say in theory about the recovered causal order if a portion of variables are intervened on?).”
>
> Thank you for raising this important point. We agree that analyzing intervention on all variables can be restrictive. Our work, however, extends to partially intervened settings as well: results relevant to your question are encapsulated in Theorem 2, Theorem 3, Theorem 4, and Lemma 6 of our paper. These demonstrate robust theoretical guarantees even when only a subset of variables is intervened upon. Specifically, these results address scenarios where each variable is included in the intervention set with a probability $\(p_{\text{int}} < 1\)$, assuming that, on average, the set $\(I\)$ of intervened variables is a subset of the set $\(V\)$ of all variables.
> Consider for example the problem of identifying the causal order in gene regulatory networks for a specific cell line. A widely used intervention mechanism in genomics is CRISPR-Cas9 gene knockout, where a library determines which genes are targeted for intervention. In practice, these knockout libraries often do not include the full set of genes in the genome; instead, they target a specific subset of genes. The choice of targeted genes varies across research labs due to differences in experimental focus or resource constraints.
> Now, consider the scenario where each lab independently designs its own knockout library. We can thus assume that the knockout are chosen approximately uniformly at random across all genes in the genome, assuming sufficient diversity in the knockout libraries. This independence across labs reflects a realistic setting where no coordinated global strategy for target selection exists. Our theoretical results provide a strong guarantee on Intersort average error on these partial intervention experiments across labs.

---

> > ### Author Response · Authors · 2024-11-23
> >
> > > “Moreover, the proposed algorithm is designed under the causal sufficiency assumption (which as far as I checked, is not clearly mentioned in the paper).”
> >
> > Thank you for your detailed feedback regarding the assumption of causal sufficiency and its mention in the original version of the paper. Your insights prompted us to re-evaluate the necessity of this assumption in our model. We are grateful to you and other reviewers for highlighting this aspect, which has led us to enhance the clarity and scope of our work. Initially, the definition of our Structural Causal Model (SCM) implicitly implied causal sufficiency. After re-examining this assumption, we have revised the manuscript to clarify that causal sufficiency is not a requirement for our proposed algorithm and theoretical guarantees.
> >
> > We have introduced a new section in our revised manuscript, titled "Latent Confounders," where we detail the applicability of our approach in scenarios with hidden confounders. By utilizing the established framework for Acyclic Directed Mixed Graphs (ADMG), we demonstrate that our methods and the corresponding theoretical guarantees remain valid even in the presence of latent confounders. In ADMGs, confounding is modeled through bidirected edges that reflect dependencies among observed variables caused by unobserved common causes. This new section improves the paper by showing that our algorithm not only is expected to perform well under the assumption of causal sufficiency but also extends its applicability to more realistic scenarios where some variables are unobserved or influenced by hidden factors.
> > We value this opportunity to refine and broaden our contribution to accommodate scenarios with partial observations, making our work relevant to a more extensive array of applications in real-world data analysis.
> >
> > >“In lines 65-68, the authors are giving the advantage of knowing the order. Did they mean that the number of candidates is divided by 4 if we know that the target gene is in the middle of the causal order?“
> >
> > When we know that the target gene lies in the middle of the causal order, the number of genes to consider on each side of the target is effectively halved, simplifying the calculation by considering ( n/2 ). Consequently, the number of potential pairs is reduced to $( C(n/2, 2) = \frac{n(n-2)}{4} )$, which represents a halving of the total candidates rather than a division by 4, as you suggested.
> > The ratio of the reduced number of candidates to the original configuration is computed as: $[ \text{Ratio} = \frac{C(n/2, 2)}{C(n, 2)} = \frac{n(n-2)/4}{n(n-1)/2} = \frac{n-2}{2(n-1)}. ]$ As $( n )$ increases, this ratio approaches $( 1/2 )$, indicating an effective reduction to half the candidates rather than a quarter.
> > We hope this clarification adequately addresses your query.
> >
> > >“In eq. (1), in the second term, why is there a coefficient of d?”
> >
> > The coefficient $(O(d))$ in the second term of the score is a technical modification intended to emphasize the influence of causal effects that are greater than the significance threshold $(\epsilon)$. To provide some intuition: the terms in the summation where the statistical distance is smaller than $(\epsilon)$ contribute negatively to the score by an order of magnitude roughly proportional to $(d)$ for each intervention. This inflation is necessary to ensure that the score strongly favors orderings where $(\pi(i) < \pi(j))$ for any pair $((i, j))$ where the distance exceeds $(\epsilon)$. By weighting the significant effects more heavily, the scoring function guides the optimization towards a causal order that better reflects meaningful causal relationships in the data.
> > In revising the presentation of this scoring method in our paper, we've included a more detailed explanation to provide better insight into this construction.
> > By including this expanded explanation, we hope to make the reasoning behind our scoring formulation more transparent and accessible to our readers.
> >
> > >“The pseudocode in Algorithm 1 is somehow useless. The most important parts (SORTRANKING and LOCACLSEARCH) are not described there.“
> >
> > We understand that the absence of detailed procedures for critical components like SORTRANKING and LOCALSEARCH may reduce its utility. To optimize the use of space in the main body of our paper, we decided to move Algorithm 1 along with the detailed steps of SORTRANKING and LOCALSEARCH to the appendix.
> >
> > >“Evaluating empirically does not imply the quality of the output of the algorithm in theory. I suggest removing the term "approximation algorithm" in line 359.”
> >
> > We have taken your suggestion and removed the term "approximation algorithm" from line 359. We appreciate your feedback highlighting that empirical evaluation and theoretical guarantees serve different purposes, and we want our terminology to reflect that distinction accurately.

---

> > > ### Author Response · Authors · 2024-11-23
> > >
> > > >“In Lemma 4, based on the chosen , it seems that the graph is disconnected with high probability. Can the authors elaborate more on this?”
> > >
> > > We indeed reference Theorem 4 rather than Lemma 4 for this context, which assumes an Erdős-Rényi model for the graph with a given probability $(p_e)$ for edge existence. We acknowledge that such a model may not accurately capture the structure of many real-world graphs due to its tendency to form disconnected components, particularly as the graph size increases.
> > > This disconnectedness indeed presents challenges for causal order recovery, primarily because it necessitates more interventions to achieve reliable estimation. In particular, disconnected graphs often require interventions on various separate components to fully understand causal influences across the entire set of variables, which is less efficient than intervening in a more densely connected graph.
> > > Despite these theoretical constraints, our formulation is intentionally conservative. It represents a sort of "worst-case" scenario regarding connectivity to ensure that our theoretical guarantees are robust even under less than ideal conditions. In practice, if a graph includes strongly connected components, it might require fewer interventions to achieve similar recovery, as intervening on a few well-chosen members of these components could significantly reduce the expected error. Therefore, our model's assumptions allow our theoretical insights to hold in broad scenarios, potentially making them more applicable to a variety of real-world systems compared to more optimistic graph models.
> > >
> > > >“In line 283, the authors argued that the expected error is growing with the order d  and they mentioned that it is a strong guarantee. I think the upper bound on the expected error is d^2. Therefore, I am not sure this is indeed a strong result”
> > >
> > > Our paper provides a theoretical upper bound for the expected error associated with our method, as detailed in Theorem 4 and Lemma 6, which indicates an upper bound of $(O(d))$. This bound reflects the expected error growth as the number of variables $(d)$ increases and represents a robust performance guarantee across different scales.
> > > We'd love to understand the reasoning behind your impression that the upper bound might be $(O(d^2))$, as this would indeed suggest a significantly weaker level of theoretical guarantee. Our derivation highlights that under the assumptions laid out the expected error grows linearly with the order of variables, ensuring the method remains practical and efficient even as the problem scale grows.
> > >
> > >  If there are any specific aspects of the derivation you feel might need further clarification, we are more than willing to dive deeper into those assumptions or steps.
> > >
> > > >“What is the computational complexity of the proposed algorithm? It is also good to compare the algorithms in terms of runtime empirically.”
> > >
> > > The computational complexity of the SORTRANKING component of our algorithm is $(\mathcal{O}(d \cdot |\mathcal{I}| \log (d \cdot |\mathcal{I}|)))$, which accounts for sorting the matrix of distances. For LOCALSEARCH, the complexity of each iteration is $(\mathcal{O}(d^{2k}))$. In our experiments, (k = 1), implying that the complexity grows quadratically with the number of variables per iteration. The number of iterations correlates with how well the initial solution approximates the optimal, but we conjecture this number to be approximately linear in $(d)$.
> > >
> > > Acknowledging the importance of empirical runtime analysis, we have added Figure 9 in Appendix F2, which compares the runtime of INTERSORT and baseline algorithms for the Neural Network data type setting. While INTERSORT shows reasonable runtimes with up to 30 variables, we do observe a significant 100-fold increase in runtime when scaling from 10 to 30 variables. This considerable rise as the input size increases highlights a limitation in scalability for much larger settings, which we have discussed within the main manuscript.

---

> > > > ### Author Response · Authors · 2024-11-23
> > > >
> > > > >“In Fig. 2, in some settings (such as NN 30 variables or GRN with a fraction of 23.33%), other methods have better performance. I think it is good to elaborate more on these cases in the experiment section.”
> > > >
> > > > Thank you for pointing out the performance differences observed in Figure 2 under certain settings, such as NN with 30 variables and GRN at a 23.33% intervention fraction. We acknowledge that INTERSORT does not always outperform other methods, particularly in these contexts.
> > > > The differences can be attributed to the type of information utilized by the methods. Baseline methods may benefit from additional information, such as functional dependencies between variables, which can enhance performance in specific scenarios. INTERSORT relies solely on observed changes in the marginal distribution of the variables due to interventions. As the fraction of observed interventions decreases, the efficacy of INTERSORT approaches that of random chance, particularly when the intervention coverage is low, such as at 23% for the GRN setting. This limitation arises because fewer interventions mean fewer directly observable changes, which diminishes the ability of the algorithm to distinguish causal ordering effectively.
> > > >
> > > >
> > > > We would be grateful if you could review these revisions and let us know before the end of the discussion period if there are any remaining issues or if further clarification is needed.

---

> > > > > ### Comment · Reviewer_R6EY · 2024-11-26
> > > > >
> > > > > I thank the authors for the detailed response.
> > > > >
> > > > > **Comparison with (Tian and Pearl 2013)**:
> > > > > I still believe that a detailed comparison between the current work and (Tian and Pearl 2013) should be done in terms of methodology and theoretical results (when $\epsilon=0$). I think the authors' claim "Our work brings two major contributions: first, we propose a score-based approach, in contrast to the rule-based algorithm of Tian & Pearl (2013)" is wrong. In Section 6 of (Tian and Pearl 2013), they took a Bayesian approach and provided the BDe score.
> > > > >
> > > > > **Other related work**: There are some more recent works like DCDI (it does not support latent confounding), or JCI (it allows latent confounding) for learning causal structure without even knowing the location of interventions. The key advantages of the current method with respect to these methods should be mentioned explicitly in the paper.
> > > > >
> > > > > **About the assumption of single interventions on all variables**: As far as I know, the number of interventions performed in a lab is limited (maybe around 100 interventions in a reasonable time). We may want to study more than 15000 genes. Therefore, we need to collect interventional data from 150 labs (hoping that there is no overlap between them)!
> > > > >
> > > > > **About the upper bound**: I forgot to write "trivial" in my sentence. What I meant is that the trivial upper bound is $d^2$ and I am not sure providing an upper bound of $d$ is a strong result.
> > > > >
> > > > > **About Lemma 4**: Why is the statement correct? "In particular, disconnected graphs often require interventions on various separate components to fully understand causal influences across the entire set of variables, which is less efficient than intervening in a more densely connected graph." To me, the most challenging cases are graphs close to being complete. The most difficult one is the complete graph which requires many interventions (see Eberhardt, Frederick, Clark Glymour, and Richard Scheines. "On the number of experiments sufficient and in the worst case necessary to identify all causal relations among n variables." arXiv preprint arXiv:1207.1389 (2012).) I think the result in very sparse graphs with many isolated nodes does not showcase the performance of the algorithm in real graphs.
> > > > >
> > > > > In summary, I think the submission requires substantial revisions. Even new results under latent confounding are added during the discussion period which needs further time for review.

---

> > > > > > ### Author Response · Authors · 2024-11-29
> > > > > >
> > > > > > We thank Reviewer R6EY for their response to our comments addressing their feedback.
> > > > > >
> > > > > > *Comparison to Tian & Peal (2013).*
> > > > > > We present a comparison to Tian & Pearl on page 2 and 3 of the revised manuscript. To reiterate, eps-interventional faithfulness (Def 1 in revised manuscript) incorporates a metric $D$ and a threshold $\epsilon$ in contrast to "influentiality" (Def. 2 in Tian & Pearl, 2013) that does not explicitly formalise a threshold or a distance metric. The inclusion of $\epsilon$ and $D$ are essential for permitting theoretical analysis, including the error bounds we derived based on the $\epsilon$-interventional faithfulness assumption, and also better reflect practical experimental settings in which experimental noise can determine the maximum threshold $\epsilon$ to which the data fulfills the eps-interventional faithfulness assumption. Error bounds for causal order derivation under realistic assumptions are crucial in practical applications because - in the general case - no ground truth to compare to exists, and it is therefore paramount to have supporting theoretical guarantees.
> > > > > >
> > > > > > > "I think the authors' claim "Our work brings two major contributions: first, we propose a score-based approach, in contrast to the rule-based algorithm of Tian & Pearl (2013)" is wrong. In Section 6 of (Tian and Pearl 2013), they took a Bayesian approach and provided the BDe score."
> > > > > >
> > > > > > Section 6 of Pearl & Tian (2013) introduces a Bayesian approach to calculate a posterior for a *causal diagram $G$ given a dataset and additional background knowledge $\xi$*. In contrast, we introduce - as described on page 2 and 3 of the revised manuscript - a score over a causal order (eq. (1)). Our score does not require the specification of a full causal diagram and does not require the additional assumptions introduced in Section 6 of Tian & Pearl (2013).
> > > > > >
> > > > > > More precisely, the Bayesian approach and the corresponding score discussed in Section 6 of their paper operate within a two-phase framework. In this framework, causal information is first derived from dynamic (interventional) data and then incorporated as prior knowledge to be combined with static (observational) data. In contrast, our approach focuses solely on the first phase of this sequential method, where the score is used to identify the best (top-scoring) causal order based on interventional data alone.
> > > > > > Importantly, our approach could naturally serve as a replacement for the first phase in their Bayesian framework, enabling the combination of interventional and observational data to derive a new Bayesian score. However, this extension lies beyond the current scope of our work, which is specifically focused on the derivation and application of the interventional score.
> > > > > >
> > > > > > It is therefore to the best of our knowledge strictly true that Pearl & Tian (2013) do not introduce a score over causal orders (instead they introduced a posterior probability for causal diagrams under additional background knowledge).
> > > > > >
> > > > > > > "As far as I know, the number of interventions performed in a lab is limited (maybe around 100 interventions in a reasonable time)."
> > > > > >
> > > > > > Modern interventional libraries often comprise gRNAs targeting thousands of single genes. Please see for example the referenced Replogle et al (2022).

---

> > > > > > > ### Author Response · Authors · 2024-11-29
> > > > > > >
> > > > > > > > "I am not sure providing an upper bound of d is a strong result."
> > > > > > >
> > > > > > > We sincerely appreciate the reviewer’s feedback and acknowledge that the derived error bound may not fully meet their desired standard. Error bounds for causal order derivation under realistic assumptions are crucial in practical applications because - in the general case - no ground truth to compare to exists, and it is therefore paramount to have supporting theoretical guarantees.  However, we would like to highlight that improving from a quadratic $O(d^2)$ error bound (as you mentioned yourself) to a linear $O(d)$ bound is still a significant advancement. While we agree that "strong" is a subjective term, this improvement represents a meaningful step in the right direction, particularly given that our work aims to balance practical algorithm design with theoretical guarantees. The derived bound complements our focus on providing both actionable methods and theoretical insights, rather than prioritizing theoretical optimality alone.
> > > > > > >
> > > > > > > To contextualize the practical relevance of this improvement, consider scenarios where the dimensionality $d$ represents a large-scale system:
> > > > > > >
> > > > > > > 1.	Genomics: In gene regulatory network discovery, $d$ might represent the number of genes. A linear error bound implies that error grows proportionally with the number of genes, while a quadratic bound would make large-scale analysis computationally prohibitive or less reliable.
> > > > > > > 2.	Sensor Networks: In a distributed system with $d$ sensors, a linear error bound ensures scalable reliability as the number of sensors increases, compared to quadratic error growth, which could overwhelm system performance.
> > > > > > > 3.	Large-scale Machine Learning: In feature selection or causality tasks with $d$ features, moving from $O(d^2)$ to $O(d)$ significantly reduces complexity in applications like explainable AI or feature attribution.
> > > > > > >
> > > > > > > We also believe that this result opens the door for future advancements, where tighter bounds may be achievable by introducing task-specific assumptions or novel theoretical frameworks. As a first step, however, demonstrating that the error bound scales linearly rather than quadratically is a substantial improvement in both theoretical understanding and practical implications.
> > > > > > >
> > > > > > >
> > > > > > > > "About Lemma 4: Why is the statement correct?  To me, the most challenging cases are graphs close to being complete. The most difficult one is the complete graph which requires many interventions”
> > > > > > >
> > > > > > > We agree that the assumption of Lemma 6 relates to a particular setting, that we consider interesting and relevant for real-world applications as many real-world settings can be assumed to be scale-free. We acknowledge that there may exist special cases where a densely connected graph may be more challenging as more interventions may be necessary to achieve the same amount of errors, especially because more edges imply a higher number of possible errors. However, we would like to point out that the presented edge case of densely / fully connected graph is already theoretically covered by our work. Indeed, a bound can be derived for this case from Thm 4, by for example taking $p_e = 1$. This also implies that for densely connected graph, the expected number of errors is also bounded by an $O(d)$ term.

---

> > > > > > > > ### Comment · Reviewer_R6EY · 2024-12-02
> > > > > > > >
> > > > > > > > I thank the authors’ response. From my understanding, the score introduced in Section 6 of (Tian and Pearl, 2013) can be used to recover a causal graph from observational and interventional data. While I agree with the authors that this score was not specifically designed to recover the causal order, it is still possible to obtain a causal order from the recovered causal graph. Consequently, it would be interesting to evaluate how algorithms like those in (Tian and Pearl, 2013) or JCI perform in the given experimental setup. I have no further comments on the other topics we discussed. Based on our discussion, I found the approach novel but believe further investigation is necessary to compare it against methods for recovering causal graphs from observational and interventional data. I have updated my score accordingly.

---

### Author Response · Authors · 2024-11-23
**Authors General Response to reviews**

We would like to express our gratitude to all reviewers for their thorough and constructive feedback on our manuscript. We are pleased that the reviewers recognized the novelty and theoretical contributions of our proposed method, INTERSORT, for inferring causal order from interventional data. The positive comments related to the soundness, presentation, and contribution of our work are greatly appreciated and affirm our effort to provide a well-founded and impactful addition to the field of causal inference.

## Highlights of Positive Feedback:
- **Originality and Contribution**: Many reviewers noted the originality of our approach in using interventional data to estimate causal order, highlighting the significance of our contributions to causal discovery (“important and highly relevant contribution” Reviewer WsAd, “several novel and theoretically grounded contributions” Reviewer mrVM).
- **Theoretical Foundation**: The theoretical underpinnings, including the development of the causal order score and the assumptions like ϵ-interventional faithfulness, as well as the theoretical guarantees even for subset of interventions, were acknowledged as sound and novel. (“ Strong theoretical guarantees and diverse empirical evaluations are presented” Reviewer PzaP)
- **Empirical Validation**: Reviewers appreciated the promising empirical results on simulated datasets, which support the effectiveness of our method compared to established baselines. (“Promising empirical results on simulated data” Reviewer mrVM, “ INTERSORT outperforms existing methods in various simulations” Reviewer WsAd)

## Summary of Main Concerns:
- **Intuition for the Score**: Most reviewers (Reviewer R6EY, Reviewer PzaP, Reviewer nPs4) requested clearer explanations of the scoring method we proposed. We have revised the manuscript to provide more intuitive explanations for Equation 1, detailing the rationale and impact of each component in the score to enhance its clarity and accessibility.
- **Section on Causal Confounders**: In response to feedback on clarifying the method’s applicability in the presence of latent confounders (Reviewer R6EY, Reviewer WsAd, Reviewer nPs4), we have added a new section discussing how our approach can be applied in these scenarios, leveraging Acyclic Directed Mixed Graphs (ADMGs) (Verma 1991, Richardson 2003) to model confounded causal relationships.
- **Comparison to related work**: To better position our contribution within the existing literature, we have expanded our discussion to include a more explicit comparison with the existing work, e.g. Tian and Pearl 2013. This comparison highlights how our score-based method differs from their rule-based algorithm and underscores the advancements and scalability offered by our approach.
- **Application on real-world data**: As suggest by Reviewer mrVM, we added a qualitative example of the output of Intersort on the real-world data of Sachs 2005, in Appendix F3, Figure 10.

All the revision to the text are colored blue in the revised manuscript. We sincerely appreciate the reviewers' valuable insights, which have significantly enhanced the quality and clarity of our manuscript. Thank you for your role in helping us refine our work.

References:

Jin Tian and Judea Pearl. Causal discovery from changes. arXiv preprint arXiv:1301.2312, 2013.

T. S. Verma. Invariant properties of causal models. Technical Report R-134, UCLA Cognitive Systems Laboratory, 1991.

Thomas Richardson. Markov properties for acyclic directed mixed graphs. Scandinavian Journal of Statistics, 30(1):145–157, 2003.

Karen Sachs, Omar Perez, Dana Pe’er, Douglas A Lauffenburger, and Garry P Nolan. Causal protein-signaling networks derived from multiparameter single-cell data. Science, 308(5721): 523–529, 2005

---

### Meta-Review · Area_Chair_NtTM · 2024-12-22

**Metareview:**

The paper introduces an approach for learning the order of a causal DAG model from single variable interventions. Reviewers found it useful, and I'm generally favourable to accept based on the framing of the algorithms being novel and for providing results others may want to build upon. But I believe scholarship could be improved, and the claims of originality better contextualized: for avoidance of doubt, the paper is not the first to address causal ordering learning from interventional data, a problem which broadly speaking is not hard under some typical assumptions. Some of my points below are mentioned by reviewer R6EY, and I'm disappointed that they are not addressed by the authors.

* I'm somewhat confused why the authors mention Tian and Pearl repeatedly as a key citation for their contribution. First, please correct the reference: Tian and Pearl's is a 2001 (!) paper from UAI. I see limited connections between the motivation of that paper and the presented paper. Tian and Pearl's main point was how to make use of natural experiments, while my interpretation is that the present contribution is motivated by experimental design. If the authors want to refer to a historical paper on using interventional data, it would make far more sense to refer to Cooper and Yoo's "Causal discovery from a mixture of experimental and observational data" (UAI, 1999). Tian and Pearl explain very clearly the connection between their paper and (last century's) literature on causal discovery from interventional data, please refer to them for details.

* As correctly mentioned in the paper, the notion of learning from pairwise orderings appears in LiNGAM-like algorithms, including DirectLiNGAM (Shimizu et al., 2011). Readers would benefit from an explicit discussion between methods at the more abstract level of being told pairwise directions, regardless of the source of data (observational vs experimental) and parametric family (linear or not). But claims such as "Reisach et al. (2021) that the causal order can be recovered by sorting the variable by variance in simulated datasets" are strangely phrased: causal order can *not* be recovered by variance in simulated datasets in general (!), only those which are either badly designed or, for some hopefully well-justified reason that is nothing but a very special case, follow a physical process where variance increases with depth of causal connection (e.g., some classes of additive error models with constant error term variance).

* Like reviewer R6EY, I'm surprised that papers like Eberhardt, Glymour and Scheines "On the Number of Experiments Sufficient and in the Worst Case Necessary to Identify All Causal Relations Among N Variables" (UAI, 2005) and the older "N-1 Experiments Suffice to Determine the Causal Relations Among N Variables" (published as a tech report in 2004) are not cited. Please do so, and explain the connections. Those papers solve a closely related (and to some extent, harder) problem than the one posed in this paper, using more classical faithfulness. In fact, a paper such as Saengkyongam and Silva "Learning Joint Nonlinear Effects from Single-variable Interventions in the Presence of Hidden Confounders" (UAI, 2020) jumps straight into the problem of causal effect identification with a given structure because it recognizes that ordering can be readily learned from the same data using methods such as simpler versions of Eberhardt et al. (as only ordering is necessary, instead of a full causal graph).

* broadly speaking, "interventional faithfulness" is not a novel concept as mentioned in the abstract, as numerous variants of it appear in papers where discovery with a mixture of interventional datasets take place, including some cited (any method which claims correctness of causal discovery with interventional data needs some notion of interventional faithfulness). The $\epsilon$ variant given here is novel to the best of my understanding, but it is worthwhile pointing to the reader that there are several $epsilon$-like variants of faithfulness, for instance the one found in Zhang and Spirtes "Strong Faithfulness and Uniform Consistency in Causal Inference" (UAI, 2003). The notion introduced in the paper is novel and worth discussing as far as I know, but it should be noticed that much work has been done in relaxing faithfulness assumptions (for instance, from the citations of Zhang and Spirtes). Tian and Pearl's influentially is a type of marginal faithfulness and has no extra degree of freedom.

If this paper is to be published at ICLR, I hope all of the above is addressed in the final version -- which I think it's doable and I'll assume good faith from the authors.

**Additional Comments On Reviewer Discussion:**

Much of the discussion refers to choice of comparisons and some of the literature. Authors engaged with the many questions broadly successfully, but did not address some of the more substantive points by R6EY on related work.

---

### Decision · Program_Chairs · 2025-01-22

Accept (Poster)